# Protein kinase C-alpha suppresses autophagy and induces neural tube defects via miR-129-2 in diabetic pregnancy

Fang Wang[1,*], Cheng Xu[1,*], E. Albert Reece[1,2], Xuezheng Li[1], Yanqing Wu[1], Christopher Harman[1], Jingwen Yu[1], Daoyin Dong[1], Cheng Wang[3], Penghua Yang[1], Jianxiang Zhong[1] & Peixin Yang[1,2]

Gene deletion-induced autophagy deficiency leads to neural tube defects (NTDs), similar to those in diabetic pregnancy. Here we report the key autophagy regulators modulated by diabetes in the murine developing neuroepithelium. Diabetes predominantly leads to exencephaly, induces neuroepithelial cell apoptosis and suppresses autophagy in the forebrain and midbrain of NTD embryos. Deleting the *Prkca* gene, which encodes PKCα, reverses diabetes-induced autophagy impairment, cellular organelle stress and apoptosis, leading to an NTD reduction. PKCα increases the expression of miR-129-2, which is a negative regulator of autophagy. miR-129-2 represses autophagy by directly targeting PGC-1α, a positive regulator for mitochondrial function, which is disturbed by maternal diabetes. PGC-1α supports neurulation by stimulating autophagy in neuroepithelial cells. These findings identify two negative autophagy regulators, PKCα and miR-129-2, which mediate the teratogenicity of hyperglycaemia leading to NTDs. We also reveal a function for PGC-1α in embryonic development through promoting autophagy and ameliorating hyperglycaemia-induced NTDs.

[1] Department of Obstetrics, Gynecology & Reproductive Sciences, University of Maryland School of Medicine, Baltimore, Maryland 21201, USA. [2] Department of Biochemistry & Molecular Biology, University of Maryland School of Medicine, Baltimore, Maryland 21201, USA. [3] Department of Obstetrics, Gynecology, Nebraska Medical Center, Omaha, Nebraska 68198, USA. * These authors contributed equally to this work. Correspondence and requests for materials should be addressed to P.Y. (email: pyang@fpi.umaryland.edu).

ncomplete neural tube closure results in neural tube defects (NTDs), severe birth defects of the central nervous system (CNS)[1]. Globally there are more than 300,000 NTD-affected pregnancies each year, and NTDs cause significant infant mortality and childhood morbidity. One out of ten babies with NTDs will die before their first year. Annual medical and surgical costs for children born with NTDs in the US are more than $200 million. Preexisting maternal diabetes significantly increases the risk of NTDs[1–4]. Even under the best preconception care, diabetic women are five times more likely to have a child with birth defects than are nondiabetic women[2]. Therefore, there is an urgent need to develop effective interventions for this disease. Mechanistic studies are the first step in revealing potential therapeutic targets for diabetes-induced NTDs.

Autophagy, a catabolic cellular organelle process, removes unwanted cellular components through double-membrane autophagosomes fused with lysosomes, and, thus, is essential for survival, differentiation, development and homeostasis[5,6]. Autophagy is required for embryonic neurulation because autophagy deficiency in Autophagy/beclin-1 regulator 1 (AMBRA1) null mutants results in massive neuroepithelial cell apoptosis and NTDs[7], reminiscent of those observed in diabetic embryopathy. However, it is unclear how maternal diabetes represses autophagy during neurulation in the developing neuroepithelium. Protein kinases such as the mammalian target of rapamycin and the AMP-activated protein kinase are among the first discovered regulators for autophagy[6]. Recent studies reported conflicting findings on the regulation of autophagy by the protein kinase C (PKC) signalling pathway. PKC activation is required for palmitic acid-induced autophagy in vitro[8]. PKC inhibitors induce autophagy whereas PKC activators attenuate starvation- or rapamycin-induced autophagy in vitro[9]. The PKC family consists of 12 isoforms that control diverse physiological and pathophysiological functions, including cell proliferation, differentiation and apoptosis[10]. Maternal diabetes-induced neuroepithelial cell apoptosis is the central mechanism underlying diabetes-induced NTDs[11–14]. These data collectively suggest that PKC activation is important in diabetic embryopathy. However, definitive molecular evidence supporting the key role of specific PKC isoforms in diabetic embryopathy is lacking, and the molecular intermediates downstream of PKC have not been characterized.

The PPAR-γ coactivator 1α (PGC-1α) regulates mitochondrial function and cell viability[15–17]. PGC-1α is abundantly present in the CNS[16]. Furthermore, overexpression of PGC-1α suppresses apoptosis[16,17], whereas reduced levels of PGC-1α sensitize cells to apoptosis[18]. Because mitochondrial dysfunction and apoptosis are interdependent and causative events in diabetic embryopathy[11,12], PGC-1α is strongly implicated in diabetes-induced NTDs. In the present study, we provided molecular evidence for the critical involvement of Prkca (PKCα) gene in diabetes-induced autophagy impairment, cellular organelle stress, apoptosis and NTDs by using Prkca knockout mice. We further revealed the downstream effectors of PKCα by demonstrating that PKCα activation

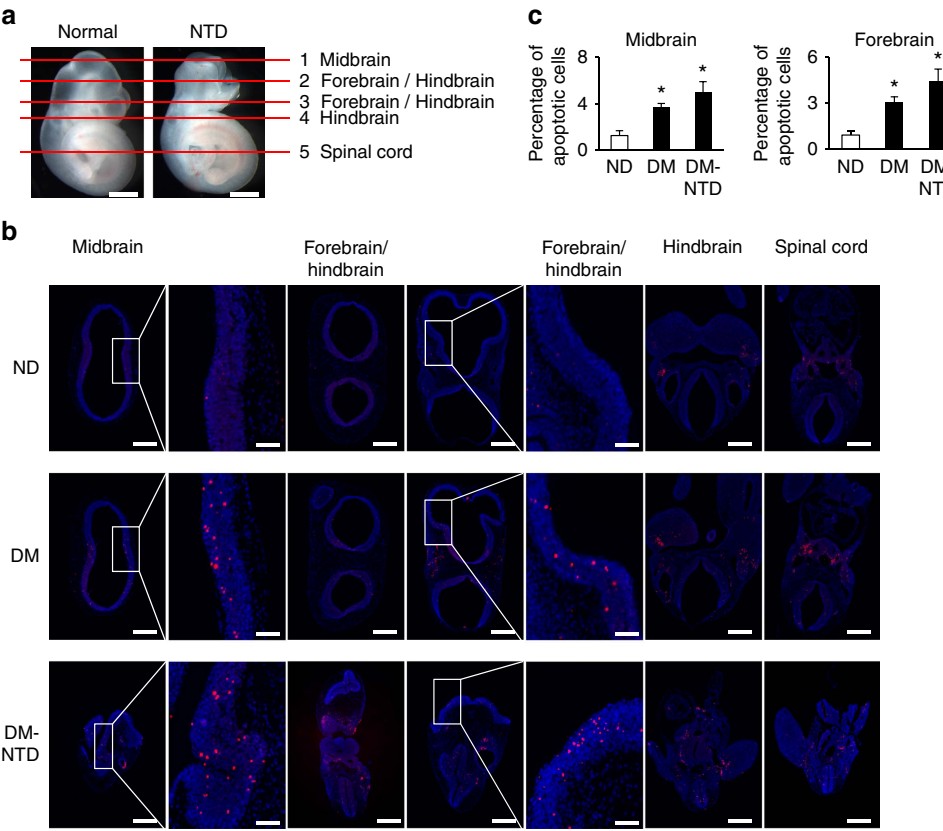

**Figure 1 | Neuroepithelial cell apoptosis in normal and NTD embryos. (a)** indicators of histological sectioning plane for **b**. E10.5 embryos were used for apoptosis detection. Red lines indicate the levels of sectioning shown on the right. For each embryo, five sections from top to bottom were chosen to detect apoptotic cells in the forebrain, midbrain, hindbrain and spinal cord were indicated in each picture. Scale bars: 300 μm. (**b**) representative TUNEL assay images showing apoptotic cells (red dots) in sections of the forebrain, midbrain, hindbrain and spinal cord. Cell nuclei were stained with DAPI (blue). Scale bars: 300 μm and 70 μm for the enlarged boxed areas. (**c**) quantification of TUNEL positive cells per section (three serial sections of each area per embryo, and three embryos from three dams were analysed) in the corresponding brain regions. ND, nondiabetic dams; DM, diabetic mellitus dams; NTD, exencephaly. *indicates significant difference (P<0.05) compared with the ND groups in one-way ANOVA followed by Tukey tests.

upregulates miR-129-2 expression. miR-129-2, in turn, repressed the expression of peroxisome proliferator-activated receptor c coactivator 1alpha (PGC-1α). PGC-1α activated autophagy both *in vivo* and *in vitro*, and PGC-1α overexpression in the developing neuroepithelium restored autophagy and cellular homeostasis, alleviating NTDs in diabetic pregnancy. Thus, our study revealed two negative autophagy regulators under maternal diabetic conditions, and provided a basis for potential therapeutic interventions for maternal diabetes-induced NTDs by targeting PKCα or miR-129-2 or enhancing PGC-1α activities.

## Results

**Apoptosis and reduced autophagy in diabetes-induced NTDs**. Maternal diabetes induced a various types of NTDs including exencephaly, craniorachischisis and spina bifida (Supplementary Fig. 1A). In addition to NTDs, microcephaly was observed (Supplementary Fig. 1A). Exencephaly was the predominant type of NTDs comprising 74.3% NTDs in the mouse model of diabetic embryopathy (Supplementary Fig. 1A). In an exencephalic embryo, serial histological sectioning revealed that failed neural tube closure occurred in the midbrain and the boundary between the forebrain and hindbrain (Supplementary Fig. 1B,C). Accordingly, excessive apoptosis was present in the midbrain and forebrain of exencephalic embryos (Fig. 1a–c). In normal embryos from diabetic dams, apoptosis was also present in the midbrain and forebrain, albeit in a less degree compared with that in exencephalic embryos (Fig. 1a–c). These findings suggest that apoptosis has to reach a threshold for NTD formation, and neuroepithelial cell apoptosis in normal embryos from diabetic dams may be related to microcephaly. Autophagy represented by LC3-GFP green puncta was diminished in the midbrain, forebrain and hindbrain of exencephalic embryos (Fig. 2a–c). There was a slight decrease of autophagy in corresponding brain areas of normal embryos from diabetic dams compared with that of normal embryos from nondiabetic dams (Fig. 2a–c). Excessive apoptosis and reduced autophagy are manifested in the brains of NTD embryos, suggesting that these events may be causal events in diabetic embryopathy. Subsequent studies were performed in neurulation stage embryos before neural tube closure to determine the causal relationship between reduced autophagy/enhanced apoptosis and NTD formation.

**Diabetes activates PKCα through oxidative stress**. Our previous study demonstrated that phosphorylated (p)-PKCα in whole embryonic lysates from diabetic dams was increased, and that overexpressing superoxide dismutase 1 (SOD1) mitigated oxidative stress and abolished this increase[19]. Here, we examined where PKCα is activated in the developing embryo in response to hyperglycaemia. We observed that the pro-apoptotic protein kinase C-alpha (PKCα) was specifically activated, along with enhanced superoxide production, in the developing neuroepithelium by diabetes (Fig. 3a,b). SOD1 overexpression abrogated diabetes-induced PKCα activation and superoxide production (Fig. 3a,b).

***Prkca* gene deletion ameliorates NTDs and restores autophagy**. We tested the functionality of PKCα activation in diabetes-induced NTDs using PKCα knockout mice[20]. Deletion of the *Prkca* gene ameliorated NTD formation, but did not affect maternal hyperglycaemia in diabetic dams (Fig. 3c,d and Supplementary Table 1). *Prkca* deletion prevented maternal diabetes-induced reduction in autophagosome numbers in neuroepithelial cells (Fig. 4a). The number of microtubule-associated protein light chain 3 (LC3)-GFP puncta, another index of autophagosome formation, was lower in neuroepithelial cells of

embryos exposed to diabetes compared with those in the nondiabetic group (Fig. 4b). The reduction of LC3-GFP puncta by diabetes was blocked by *Prkca* deletion (Fig. 4b).The decrease in conversion of LC3-I to LC3-II, a biochemical parameter for autophagy, was observed in embryos under diabetic conditions, and this decrease was abrogated in the absence of PKCα (Fig. 4c). The downregulation of autophagy promoting factors, and the upregulation of autophagy negative regulators, by diabetes were also reversed by *Prkca* deletion (Supplementary Fig. 2A).

To further examine whether PKCα suppresses autophagy, we utilized the C17.2 neural stem cell line[21,22], which exhibits high basal autophagic activities as those in neuroepithelial cells of the developing neuroepithelium[12]. Using Cyto-ID staining, we observed that the constitutively active PKCα (caPKCα) mutant and a PKCα pharmacological activator mimicked the inhibition of autophagy by high glucose (Fig. 4d,e). In contrast, PKCα siRNA knockdown blunted high glucose-induced autophagy reduction (Fig. 4f and Supplementary Fig. 2B,C).

***Prkca* deletion relieves cellular organelle stress**. Because autophagy maintains cellular homeostasis by removing and recycling nutrients from damaged mitochondria and the endoplasmic reticulum (ER)[23], we next examined whether *Prkca* deletion-restored autophagy could reduce diabetes-induced defective mitochondria[12,24] and stressed ER[13]. Indeed, *Prkca* deletion reduced the number of defective mitochondria in neuroepithelial cells of embryos under maternal diabetic conditions (Fig. 5a, Supplementary Fig. 2D). Maternal diabetes-induced mitochondrial translocation of the pro-apoptotic Bcl-2 family members, Bak, Bax, Puma and Bim, was inhibited by *Prkca* deletion (Fig. 5b–e). Other mitochondrial dysfunction markers, phosphorylation of Bad and cleaved Bid (tBid) under diabetic conditions, were also diminished by *Prkca* deletion (Supplementary Fig. 2E,F). These results support the hypothesis that maternal diabetes-induced mitochondrial dysfunction is prevented by deleting the *Prkca* gene.

The major function of the ER is to post-translationally modify and properly fold the newly synthesized proteins into dimensional structures. Accumulation of misfolded proteins triggers the unfolded protein response (UPR) and ER stress[25]. The activation of the major UPR sensors, IRE1α and PERK, mediates the pro-apoptotic effect of ER stress[25]. ER stress is a causal event in the induction of diabetic embryopathy because inhibition of ER stress by 4-phenylbutyric acid blocks high glucose-induced NTD formation[13]. Autophagy can resolve ER stress[12,26]. Because *Prkca* deletion reversed diabetes-repressed autophagy, we sought to determine whether *Prkca* deletion inhibits diabetes-induced ER stress. Maternal diabetes-induced phosphorylation of IRE1α and PERK, and their downstream effectors, CHOP upregulation and eIF2α phosphorylation, X-box binding protein 1 (XBP1) mRNA cleavage and upregulation of ER chaperone genes were abrogated by *Prkca* deletion (Fig. 6a,b and Supplementary Fig. 3).

Excess apoptosis in embryonic neuroepithelial cells accounts for NTD induction in diabetic embryopathy[11–13,27–30]. Because *Prkca* deletion inhibited the two pro-apoptotic responses, mitochondrial dysfunction and ER stress, we hypothesized that deleting *Prkca* would block neuroepithelial cell apoptosis. To test this hypothesis, caspase cleavage and the percentage of apoptotic cells were measured. Maternal diabetes-induced caspase 3, 8 and neuroepithelial cell apoptosis were diminished by *Prkca* deletion (Fig. 6c–e).

**PKCα upregulates miR-129-2 that silences PGC-1α expression**. Next, we looked into how PKCα mediates the adverse effects of

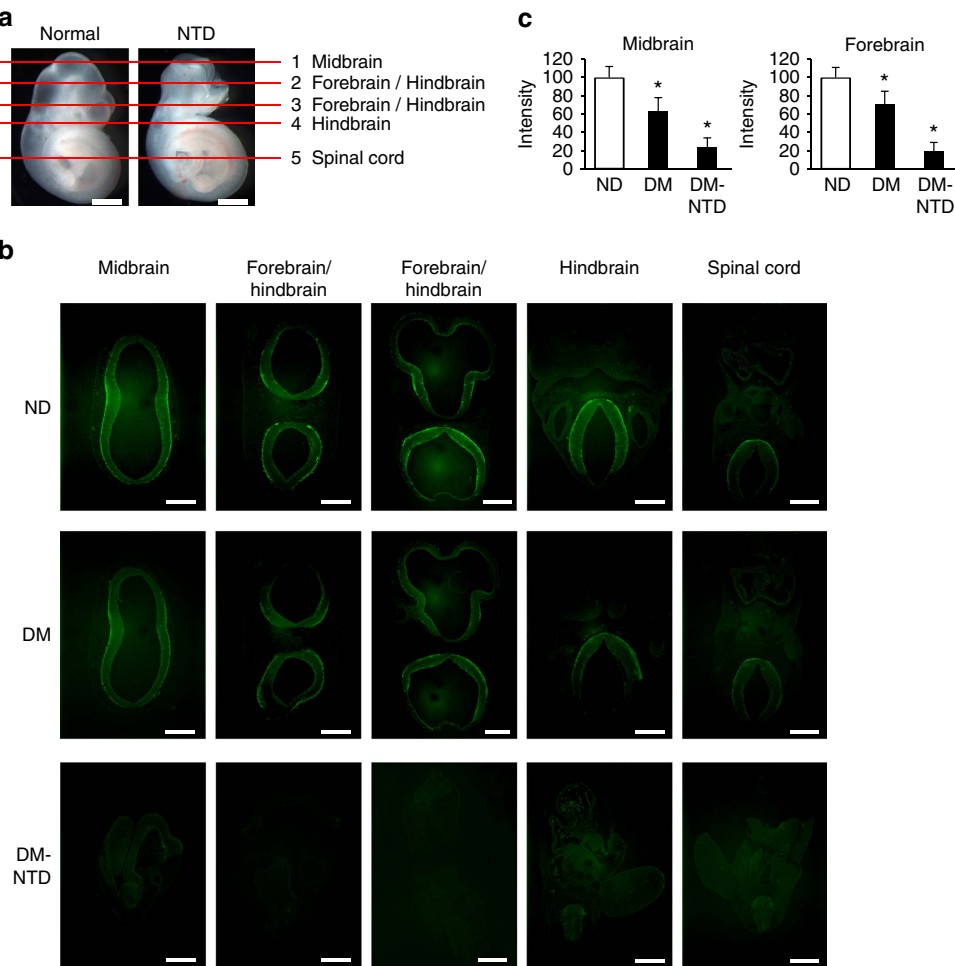

**Figure 2 | Autophagy in the developing neuroepithelia of normal and NTD embryos. (a)** indicators of histological sectioning plane for **b**. Red lines indicate the levels of sections shown below. For each embryo, five sections from top to bottom were chosen to detect autophagy levels, forebrain, midbrain, hindbrain and spinal cord were indicated in each picture. Scale bars: 300 μm. (**b**) representative images showing autophagy levels (green LC3-GFP puncta) in sections of the forebrain, midbrain, hindbrain and spinal cord. E10.5 embryos from nondiabetic (ND) and diabetic mellitus (DM) females mated with GFP-LC3 transgenic males were used for autophagy examination. Scale bars: 300 μm. Individual green LC3-GFP puncta can be clearly visualized in high magnification images in Fig. 4. (**c**) quantification of relative autophagy levels by fluorescence intensity using the ImageJ software in midbrain and forebrain (three serial sections of each area per embryo, and three embryos from three dams were analysed) in the corresponding brain regions. ND, nondiabetic embryos; DM, diabetic mellitus embryos; NTD, exencephaly. *indicates significant difference ($P < 0.05$) compared with the ND groups in one-way ANOVA followed by Tukey tests.

maternal diabetes leading to autophagy impairment and cellular organelle stress. PGC-1α, a member of the peroxisome proliferator-activated receptor c coactivator 1 (PGC1) family of transcriptional coactivators, is a positive regulator of mitochondrial biogenesis and function, and plays an important role in many metabolic processes[31]. PGC-1α is also implicated in autophagy induction by exercise and fibre type conversion in muscle[32,33]. We found that both PGC-1α mRNA and protein abundance were significantly downregulated by maternal diabetes, and were restored upon the removal of the *Prkca* gene (Fig. 7a,b). Silencing PKCα by siRNA reverted high glucose-suppressed PGC-1α expression (Fig. 7c), whereas the caPKCα mutant simulated the inhibitory effect of high glucose on PGC-1α expression in C17.2 cells (Fig. 7d). These results support our hypothesis that PKCα mediates the inhibitory effect of high glucose on PGC-1α expression.

To investigate the mechanism whereby PKCα suppresses PGC-1α expression, we focused on miRNAs because it has been demonstrated that PKCα regulates miRNA expression[34]. Additionally, there is a drastic miRNA profiling change during mouse neurulation[35], and miRNAs are implicated in human NTD formation[36]. microRNAs are non-coding small RNAs that repress gene expression by either degrading target mRNAs or blocking translation or both[37]. A miRNA target prediction algorithm (www.microrna.org) revealed miR-129-2 as a potential negative regulator of PGC-1α expression, suggesting a role of miR-129-2 in diabetic embryopathy. To determine whether miR-129-2 targets PGC-1α mRNA, we performed an RNA pull-down assay using biotin-labelled miR-129-2. PGC-1α mRNA was enriched about sevenfold in biotin-labelled miR-129-2 (Supplementary Fig. 4A). The miR-129-2 mimic repressed the luciferase reporter activities driven by the 3′-untranslated region (UTR) but not the coding region or the binding site mutated 3′-UTR of PGC-1α mRNA (Supplementary Fig. 4B).

The miR-129-2 mimic mimicked high glucose to downregulate PGC-1α in cultured C17.2 cells (Fig. 7e,f and Supplementary Fig. 4C,D).The miR-129-2 inhibitor, which forms a duplex with endogenous miR-129-2 and inactivates it, slightly increased PGC-1α expression at 50 nM but reduced PGC-1α expression at high concentrations, probably due to cell toxicity (Supplementary Fig. 4E,F). Under normal glucose condition, PGC-1α expression in cultured C17.2 cells reached a plateau that

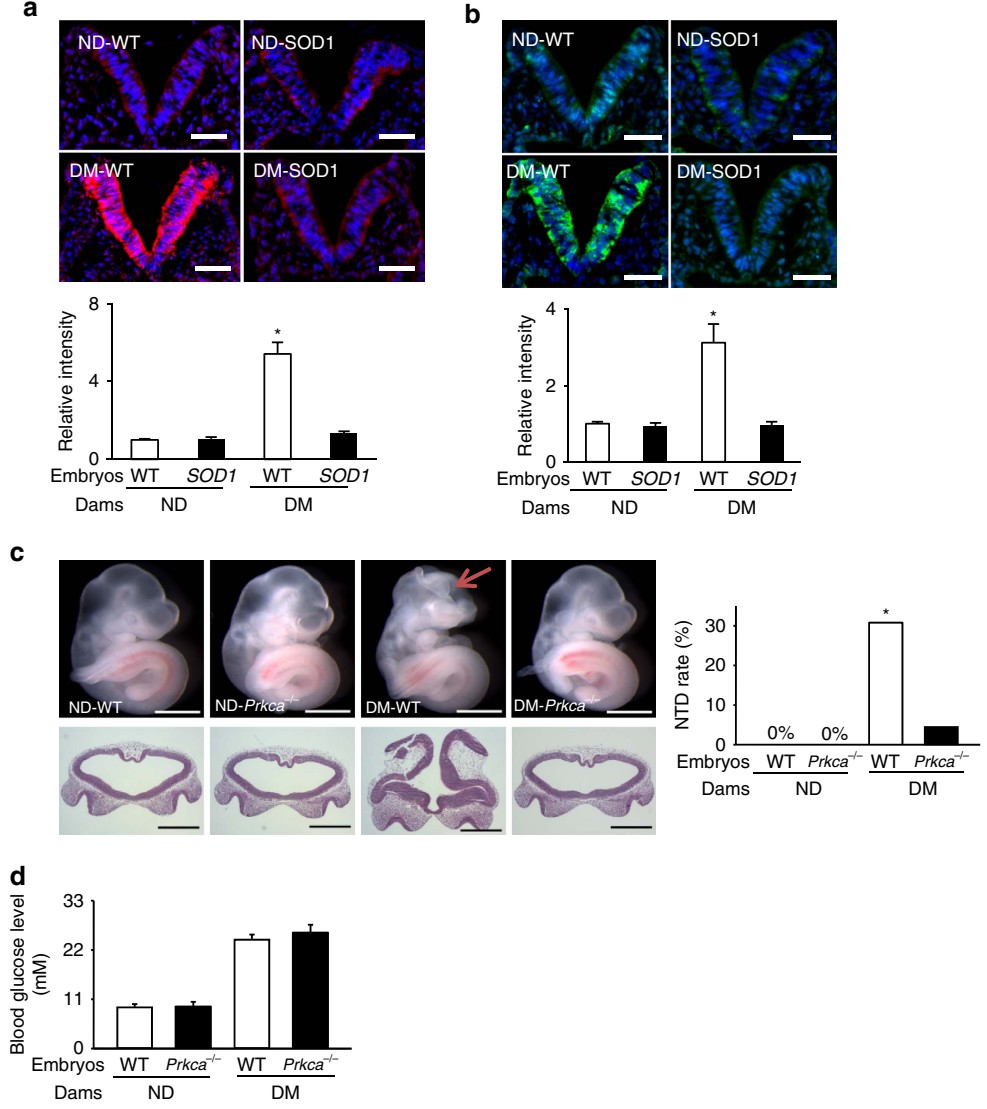

**Figure 3 | SOD1 blocks diabetes-induced ROS and *Prkca* deletion ameliorates diabetes-induced NTDs.** Dihydroethidium (DHE) staining and quantification for superoxide (**a**) and immunostaining of phosphorylated PKCα (**b**) in the E8.75 neuroepithelium (the V shape structure) of embryonic sections. All cell nuclei were stained with DAPI (blue). Scale bars = 70 μm. In **a,b** three embryos from three different dams in each group were analysed and quantified by fluorescence intensity using the ImageJ software (*N* = 3). (**c**) Morphology of E10.5 embryos from ND WT, ND *Prkca*$^{-/-}$, DM WT and DM *Prkca*$^{-/-}$ mice, and NTD rates in E10.5 embryos. Scale bars = 1 mm. The red arrow indicates exencephaly. Lower panel: frontal sections of embryos in the upper panel showing an open neural tube. (**d**) Blood glucose concentrations. *N* for **c,d**, the numbers (*N*) of embryos per condition and statistical analyses are indicated in Supplementary Table 1. DM, diabetic mellitus; ND, nondiabetic; WT, wild-type. *indicates significant difference (*P* < 0.05) compared with the ND groups in one-way ANOVA followed by Tukey tests (**a,b**) and chi-square tests (**c**).

could not be further increased by the miR-129-2 inhibitor (Supplementary Fig. 4E,F). Nevertheless, 50 nM miR-129-2 inhibitor abolished high glucose-inhibited PGC-1α protein expression and partially restored PGC-1α mRNA expression in cultured C17.2 cells (Fig. 7g,h). Thus, miR-129-2 represses PGC-1α expression by degrading mRNA and inhibiting translation. An RNA-immunoprecipitation (RIP) assay was used to assess whether PGC-1α mRNA and miR-129-2 are co-enriched in the RNA-induced silencing complex. Maternal diabetes enhanced the co-presence of PGC-1α miRNA and miR-129-2 in RNA-induced silencing complex (Fig. 7i,j), suggesting that PGC-1α is a miR-129-2 target *in vivo*. Both *in vitro* and *in vivo* data support the hypothesis that maternal diabetes- or high glucose *in vitro*-induced miR-129-2 represses PGC-1α expression.

miR-129-2 is downstream of PKCα because caPKCα simulated high glucose in upregulating miR-129-2 (Fig. 7k and Supplementary Fig. 4G), and siRNA knockdown of PKCα, or *Prkca* gene deletion blocked high glucose- or diabetes-increased miR-129-2 expression, respectively (Fig. 7l,m).

**PGC-1α overexpression reverses diabetes-inhibited autophagy.** To explore the functional consequence of PGC-1α upregulation, we generated a transgenic mouse line in which *PPARGC1A* gene, along with GFP expression, were driven by the promoter of nestin, a neural stem cell marker (Fig. 8a). Consistent with prior reports[11,38,39] that the nestin promoter-driven transgene expression occurs at E8.0 onwards, GFP signals were robustly present in the E8.5 neuroepithelia of *PPARGC1A* transgene positive (*PPARGC1A*$^+$) embryos but not wild-type (WT) embryos (Fig. 8a). Maternal diabetes-reduced autophagosome formation was restored in neuroepithelial cells of *PPARGC1A*$^+$

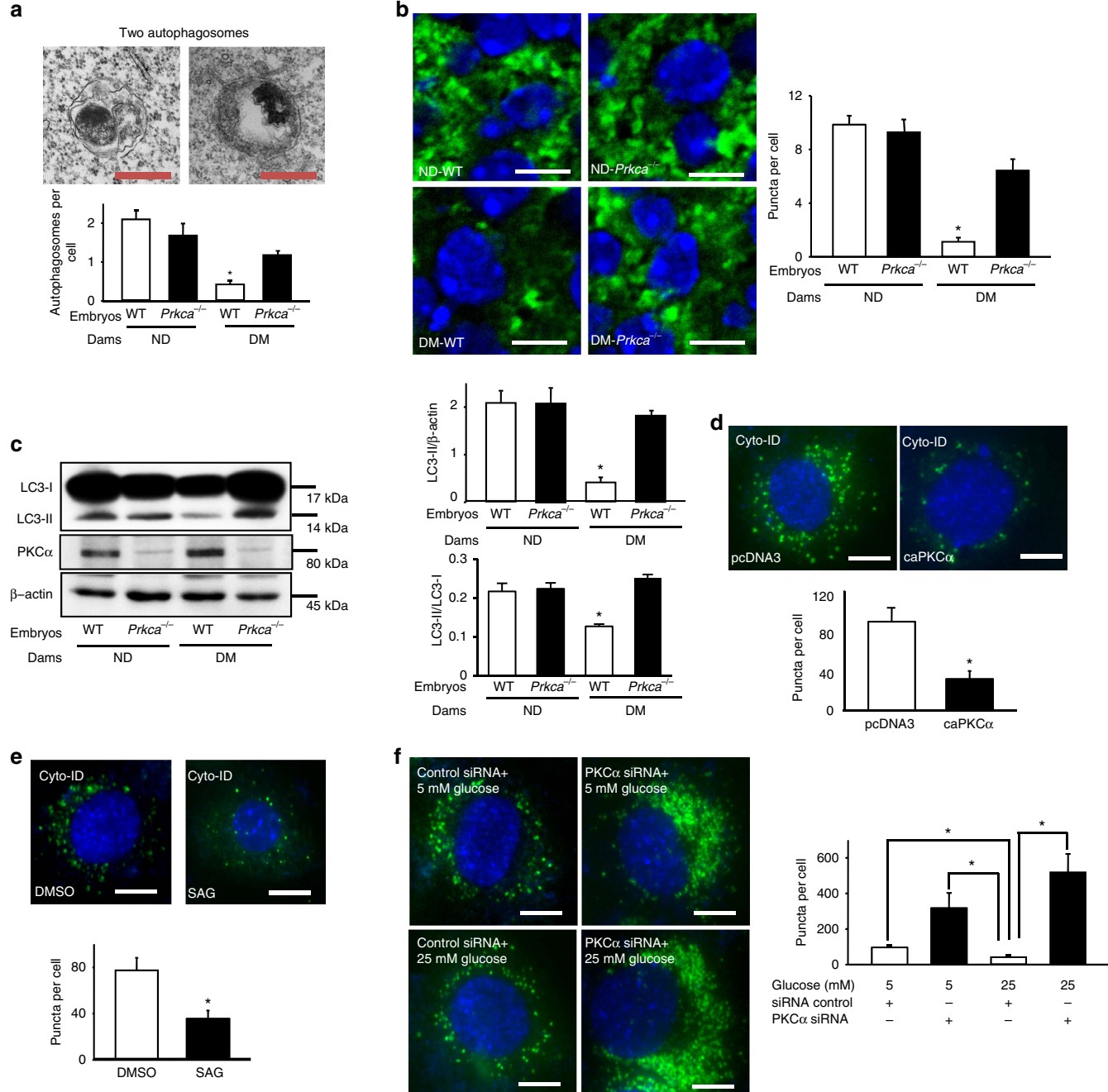

**Figure 4 | *Prkca* deletion reverses maternal diabetes-induced autophagy impairment.** (**a**) Two autophagosomes in E8.75 neuroepithelial cells are depicted in EM micrographs. Scale bars = 100 µm. EM pictures were taken with an electron microscope (model: Joel JEM-1200EX) under 12K resolution. Each image covered 9.7 µm² areas. The numbers of autophagosome and cell on each image were counted to get the value of autophagosome per cell. Five images of each embryo and three embryos from different mothers were quantified for each group. Total 86 autophagosomes from the four groups were counted. (**b**) Quantitative data of GFP-LC3 punctate foci (green dots) with a diameter greater or equal to 20 pixels in E8.75 neuroepithelial cells of embryonic sections. Cell nuclei were stained with DAPI (blue). Scale bars = 3.5 µm. (**c**) LC3-II abundance in E8.75 embryos. All experiments were performed using three embryos from three different dams ($N = 3$) and graphs in **a**–**c** showed summaries of all data. (**d**–**f**) Representative images and quantification of autophagosomes in C17.2 neural stem cells using Cyto-ID staining (Green). caPKCα: constitutively active PKCα. pcDNA3: backbone vector for caPKCα. SAG (1-Stearoyl-2-arachidonoyl-sn-glycerol): a pharmacological PKCα activator. DMSO: a vehicle control for SAG. Control: 5 mM glucose; High glucose (HG): 25 mM glucose. In **f** PKCα siRNA knockdown reversed high glucose-suppressed autophagy. In **d**–**f** experiments were repeated three times ($N = 3$). Scale bars = 15 µm. *indicates significant difference compared with other group or groups in one-way ANOVA followed by Tukey tests (**a**–**c**,**f**) and t-tests (**d**,**e**). $P < 0.05$.

embryos, which had a comparable number of autophagosomes to that of neuroepithelial cells of WT embryos under nondiabetic conditions (Fig. 8b). The reduction in converting LC3-I to LC3-II by diabetes was abrogated in *PPARGC1A*⁺ embryos (Fig. 8c). Diminished LC3-GFP puncta in neuroepithelial cells under diabetic conditions was reverted by the *PPARGC1A* transgene expression (Fig. 8d). Additionally, maternal diabetes-altered autophagy-related gene expression was corrected in *PPARGC1A*⁺ embryos (Supplementary Fig. 5A).

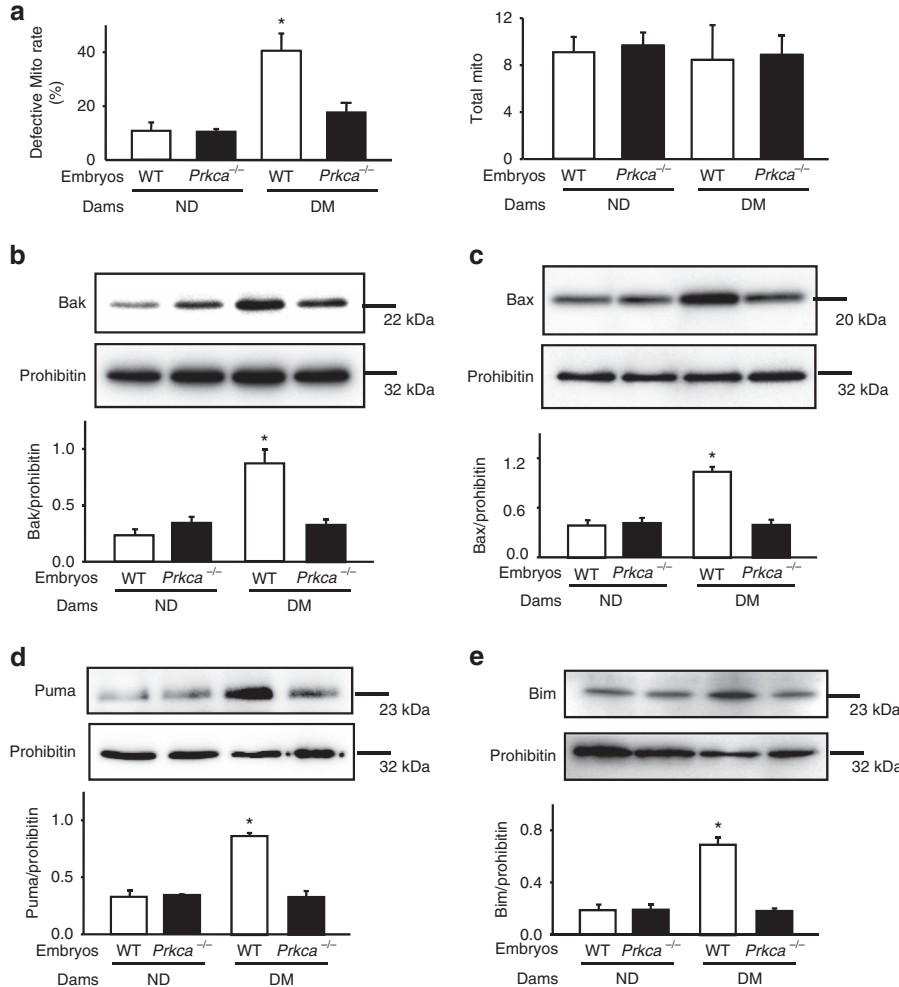

**Figure 5 | Deletion of the *Prkca* gene restores mitochondrial function.** (**a**) Defective mitochondria (Mito) rates and total number of mitochondria in embryonic sections. Defective mitochondria rate = number of defective mitochondria divided by total number of mitochondria per image area (image size: 9.43 μm²) in neuroepithelial cells based on the images of defective and normal mitochondria in supplementary fig. 2. Neuroepithelia from three embryos (N = 3) derived from different dams were used. Three serial sections per embryo were analysed. The effect of *Prkca* gene deletion on protein abundance of Bak (**b**), Bax (**c**), Puma (**d**) and Bim (**e**) in isolated mitochondria of E8.75 embryos. Bar graphs for protein abundance were quantitative data from three independent experiments. *indicates significant difference compared with other groups in one-way ANOVA followed by Tukey tests. P < 0.05.

**PGC-1α directly activates autophagy**. To confirm the autophagy-promoting effect of PGC-1α, the LC3-GFP autophagy reporter cell line was used[40]. PGC-1α transfected cells displayed robust LC3-GFP puncta, autophagosomes (Supplementary Fig. 5B). In a time-course experiment, LC3-GFP puncta as well as mitochondrial mass were induced as early as only 6 h after ectopic PGC-1α overexpression (Fig. 8e), suggesting that autophagy induction by PGC-1α overexpression is not a secondary effect of mitochondrial biogenesis. Knocking down PGC-1α using siRNA significantly attenuated autophagosome formation (Fig. 8f and Supplementary Fig. 5C,D), whereas ectopic PGC-1α overexpression inhibited high glucose-suppressed autophagy (Supplementary Fig. 5E). Therefore, PGC-1α is an autophagy promoting factor.

**PGC-1α overexpression prevents diabetes-induced NTDs**. Autophagy is essential for neural tube closure[7,12], and *PPARGC1A* overexpression restores autophagy that is suppressed by diabetes. To test the effect of *PPARGC1A* overexpression on NTD formation in diabetic pregnancies, *PPARGC1A* transgenic males mated with nondiabetic or

diabetic WT females to produce WT and *PPARGC1A*⁺ embryos under the same maternal environment. Under nondiabetic conditions, *PPARGC1A* overexpression did not affect blood glucose levels and NTD rate (Fig. 8g,h and Supplementary Table 2). Only 2.2% of *PPARGC1A*⁺ embryos under diabetic conditions exhibited NTDs, which was significantly lower compared to the 22.7% NTD rate of their WT littermates under diabetic conditions (Fig. 8h and Supplementary Table 2). The NTD rate in *PPARGC1A*⁺ embryos from diabetic dams was comparable to those of WT and *PPARGC1A*⁺ embryos from nondiabetic parents (Fig. 8h and Supplementary Table 2). Thus, similar to the findings in *Prkca* gene deletion settings, *PPARGC1A* overexpression ameliorates diabetes-induced NTD formation.

To delineate the mechanism underlying the beneficial effect of *PPARGC1A* overexpression on NTD prevention, indices of mitochondrial dysfunction, ER stress and apoptosis were assessed. Diabetes-increased number of defective mitochondria and diabetes-repressed mitochondrial gene expression were reverted by *PPARGC1A* overexpression (Fig. 9a,b). Diabetes-increased p-PERK, p-eIF2α, p-IRE1α and CHOP abundance, ER chaperone gene expression and XBP1 cleavage were abrogated in

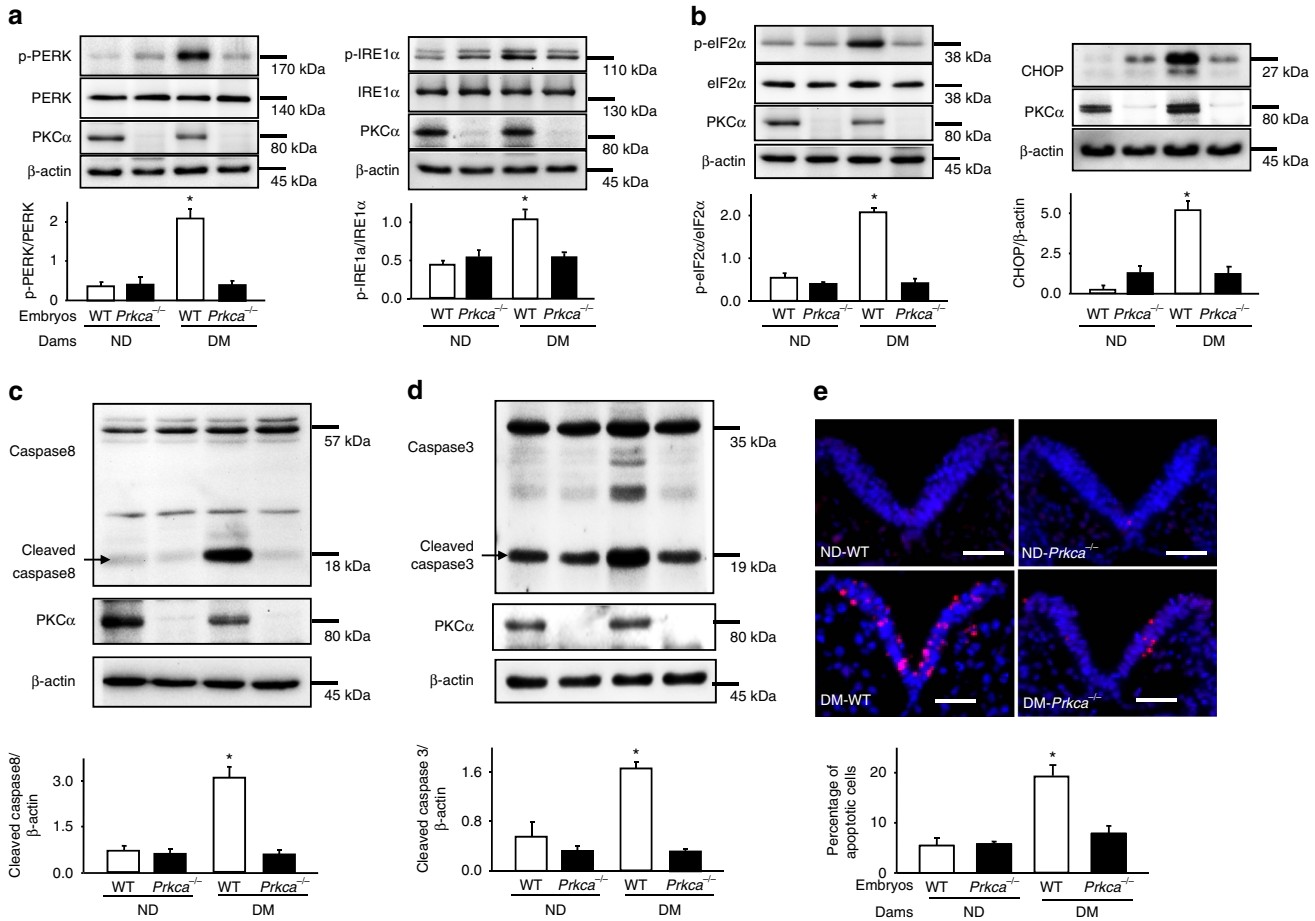

**Figure 6 | *Prkca* gene deletion restores cellular homeostasis and prevents apoptosis.** Protein abundance of phosphorylated (p-) PERK and p-IRE1α (**a**); p-eIF2α and CHOP (**b**); cleaved caspase 8 (**c**) and caspase 3 (**d**) in E8.75 embryos. Bar graphs for protein abundance were quantitative data from three independent experiments. Representative images of the TUNEL assays in E8.75 embryos (**e**). Apoptotic cells were labelled in red and nuclei were labelled in blue by DAPI. The dense blue V shape areas are the neural plate. The level of the body axis was at hindbrain level. The bar graph showed the quantification of apoptotic cell number. Experiments were repeated using five embryos ($N = 5$) from different dams and five images were obtained from each embryo. Scale bars = 70 μm. DM, diabetic mellitus; ND, nondiabetic; WT, wild-type;. *indicates significant difference compared with other groups in one-way ANOVA followed by Tukey tests. $P < 0.05$.

$PPARGC1A^+$ embryos (Fig. 9c and Supplementary Fig. 6A,B,C). *PPARGC1A* overexpression blocked diabetes-induced caspase 3 and 8 cleavage, and neuroepithelial cell apoptosis (Fig. 9d–f).

Taken together, maternal diabetes-induced PKCα activation increased miR-129-2 expression, which repressed PGC-1α expression by degrading its mRNA and inhibiting translation (Fig. 9g). Either *Prkca* deletion or *PPARGC1A* overexpression restored diabetes-suppressed autophagy and cellular homeostasis, and thus prevented NTD formation (Fig. 9g). These findings unravel a potential new kinase pathway in autophagy regulation and provide the mechanistic basis for targeting PKCα, miR-129-2 and PGC-1α for intervention for maternal diabetes-induced NTDs (Fig. 9g).

## Discussion

Autophagy maintains cellular homeostasis and, thus, promotes cell survival during embryonic development[7]. Autophagy is essential for embryonic development because deletion of the autophagy-related gene 5 (Atg5) results in early embryonic lethality at the four-cell to eight-cell stages[41], and deletion of Beclin1, a key component of the autophagy initiating complex, causes embryonic lethality at E7.5 (ref. 42). At the whole organism level, two waves of massive autophagy in early lives of mice occur at the time of fertilization and the early

neonatal period[23]. At specific tissue levels, autophagy plays an important role for cell differentiation and patterning of many tissues[23]. The developing neuroepithelium has a high level of autophagy during the period of neurulation[7,12]. This high level of autophagy is essential for neurulation or neural tube closure[7,12]. Maternal diabetes suppresses autophagy in neuroepithelial cells leading to NTD formation[12]. The present study reveals critical molecular intermediates of a signalling pathway that mediates the inhibitory effect of maternal diabetes on autophagy in the developing neuroepithelium.

It has been shown that PKC inhibitors trigger autophagy whereas PKC activators suppress starvation- or rapamycin-induced autophagy[9]. In agreement with the inhibitory effect of PKC on autophagy[9], our *in vivo* and *in vitro* evidence supports the hypothesis that PKCα is a negative regulator of autophagy and mediates the inhibitory effect of maternal diabetes on autophagy. High basal autophagic activities in neuroepithelial cells are likely required for cell survival because autophagy impairment induces neuroepithelial cell apoptosis in the developing neuroepithelium[7,12]. PKCα-suppressed autophagy under diabetic conditions causes cellular homeostatic imbalance by in favour of pro-apoptotic events. Indeed, *Prkca* gene deletion abolishes pro-apoptotic events including mitochondrial dysfunction, UPR and ER stress. Not surprisingly, removing PKCα ultimately abrogates maternal diabetes-induced neuroepithelial cell apoptosis and

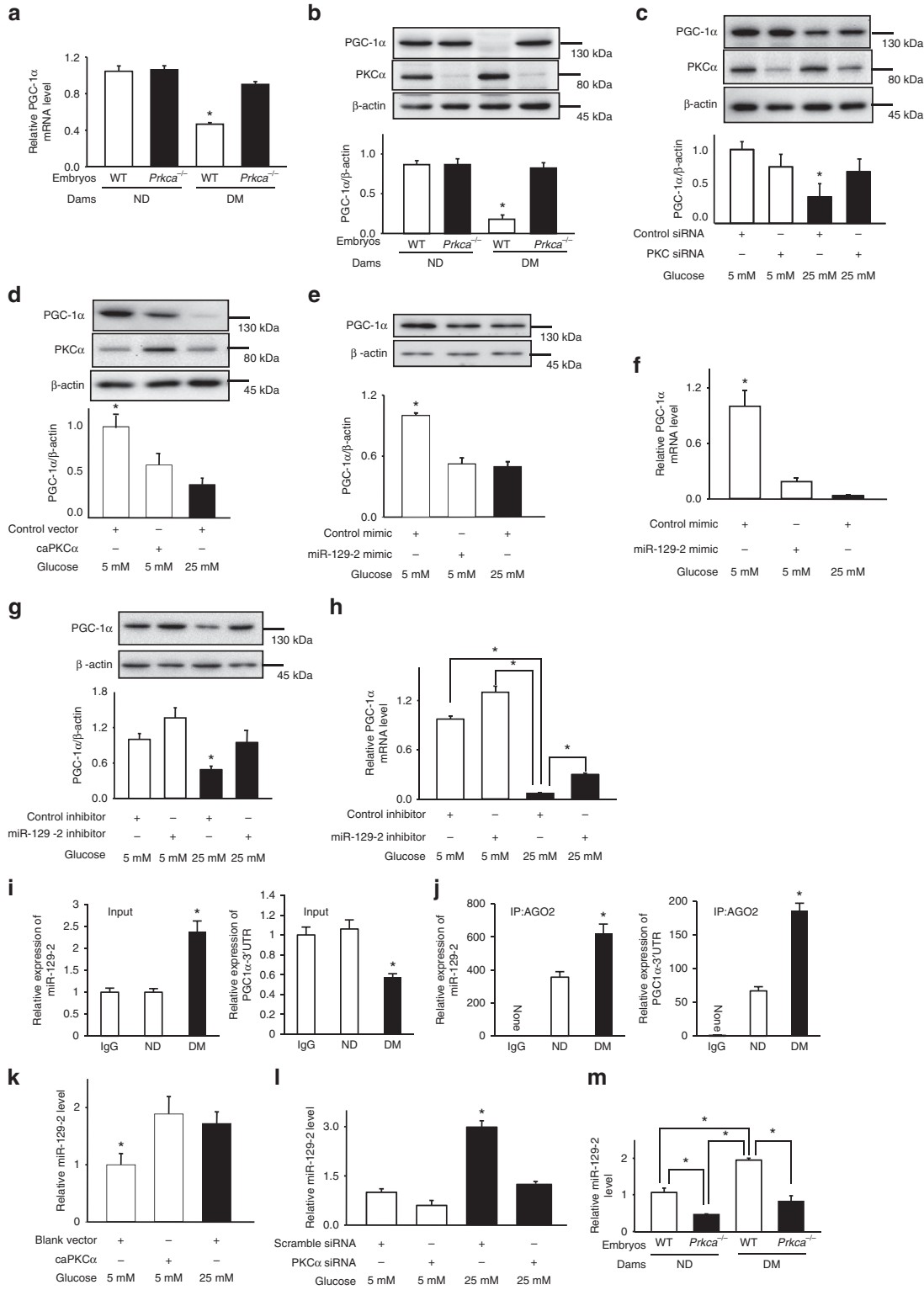

**Figure 7 | PKCα mediates the inhibitory effect of diabetes on PGC-1α through miR-129-2 upregulation.** The effect of *Prkca* deletion (PKCα[−/−]) on *PPARGC1A* (PGC-1α) mRNA (**a**) and PGC-1α protein abundance (**b**) in E8.75 embryos ($N=3$) from different dams. DM, diabetic mellitus; ND, nondiabetic; WT, wild-type. PGC-1α protein abundance in neural stem cell cultures influenced by PKCα siRNA knockdown (**c**) and constitutively active PKCα (caPKCα) (**d**). PGC-1α protein and mRNA abundance affected by the miR-129-2 mimic (**e**,**f**) and the miR-129-2 inhibitor (**g**,**h**) in C17.2 neural stem cells. (**i**,**j**) Levels of miR-129-2 and PGC-1α mRNA in an RNA immunoprecipitation assay. The RNA-induced silencing complex (RISC) in E8.75 embryos was pulled down by the AGO2 antibody, and then the levels of miR-129-2 and PGC-1α mRNA was detected in input and immunoprecipitates. caPKCα mimicked the stimulatory effect of high glucose (25 mM) on miR-129-2 expression (**k**). PKCα siRNA knockdown blocked high glucose-induced miR-129-2 expression (**l**). In **c**–**l** experiments were preformed independently three times ($N=3$). The effect of *Prkca* deletion on miR-129-2 abundance (**m**) using three embryos ($N=3$) from different dams. In **i**,**j** three litters ($N=3$) of each group were used. * indicates significant difference compared with other groups in one-way ANOVA followed by Tukey tests. $P < 0.05$.

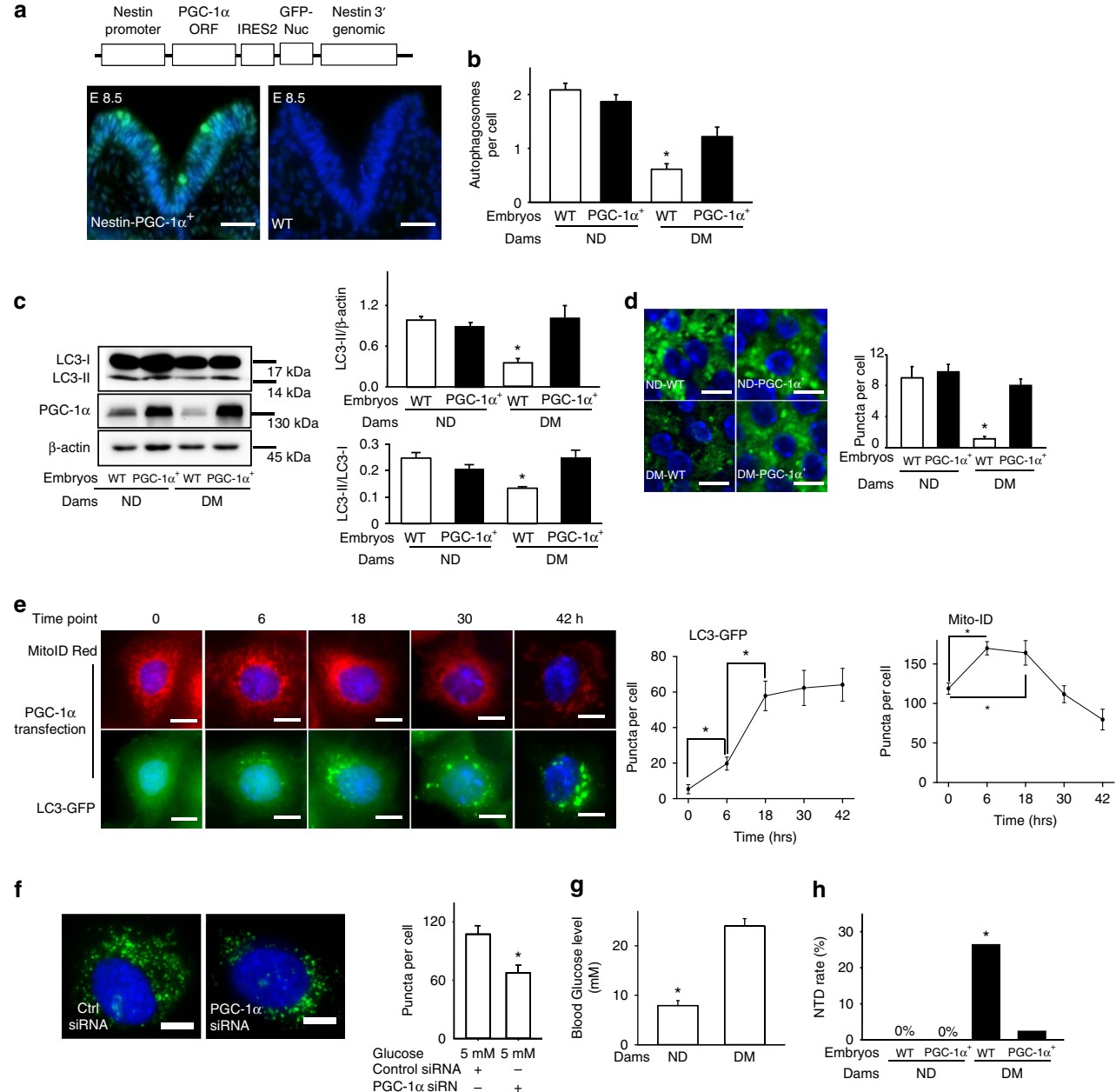

**Figure 8 | PPARGC1A overexpression restores autophagy and suppresses NTD formation.** (**a**) The transgene construct for the *PPARGC1A* (PGC-1α) transgenic mouse line, and green signals of GFP protein in the V shape neural plate of an E8.5 *PPARGC1A*⁺ embryo. The level of the body axis was at hindbrain level. Scale bars: 70 µm. (**b**) Autophagosome numbers in E8.75 neuroepithelial cells of embryonic sections. Five images of each embryo and three embryos from different mothers were quantified for each group. Total 81 autophagosomes from the four groups were counted. (**c**) LC3-II abundance in E8.75 embryos. (**d**) Quantitative data of GFP-LC3 punctate foci (green dots) in E8.75 neuroepithelial cells. Cell nuclei were stained with DAPI (blue). Scale bars = 3.5 µm. All experiments were performed independently three times (*N* = 3) using embryos from different dams, and graphs in **b–d** showed summaries of all data. (**e**) Representative images and quantitative data of GFP-LC3 punctate foci (green dots) with a diameter greater or equal to 20 pixels, and mitochondrial mass (Mito-ID red signals) in GFP-LC3 HeLa reporter cells. Scale bars = 15 µm. *indicates significant difference compared to the previous time point. (**f**) Representative images of Cyto-ID staining puncta, which represented autophagosomes, and quantification of puncta. Scale bars = 3.5 µm. In **e,f** experiments were performed three times (*N* = 3). (**g**) Blood glucose concentrations from nondiabetic (ND) and diabetic mellitus (DM) mated with PGC-1α transgenic males. (**h**) NTD rates in E10.5 embryos. *N* for **g,h** was indicated in Supplementary Table 2. One-way ANOVA followed by Tukey tests (**b–e**), *t*-tests (**f,g**) and chi-square tests (**h**) were used. *$P < 0.05$.

consequent NTD formation. Our previous studies have demonstrated that besides PKCα, maternal diabetes also activates PKCβII and PKCδ[19,43]. Specific inhibitors of PKCβII and PKCδ reduced high glucose *in vitro*-induced NTDs[43,44]. These findings suggest that these three PKC isoforms have distinctive roles in the aetiology of diabetic embryopathy. A recent study suggests that maternal diabetes-induced defects in mesoderm formation and the

primitive streak cause NTD formation in later stages[45]. Future studies may aim to reveal whether PKCα is activated in the mesoderm and the primitive streak and determine whether deleting the *Prkca* gene specifically in mesoderm lineage reduces diabetes-induced structural birth defects.

Although PKC directly phosphorylates LC3, its inhibition on autophagy is independent of LC3 phosphorylation[9], suggesting

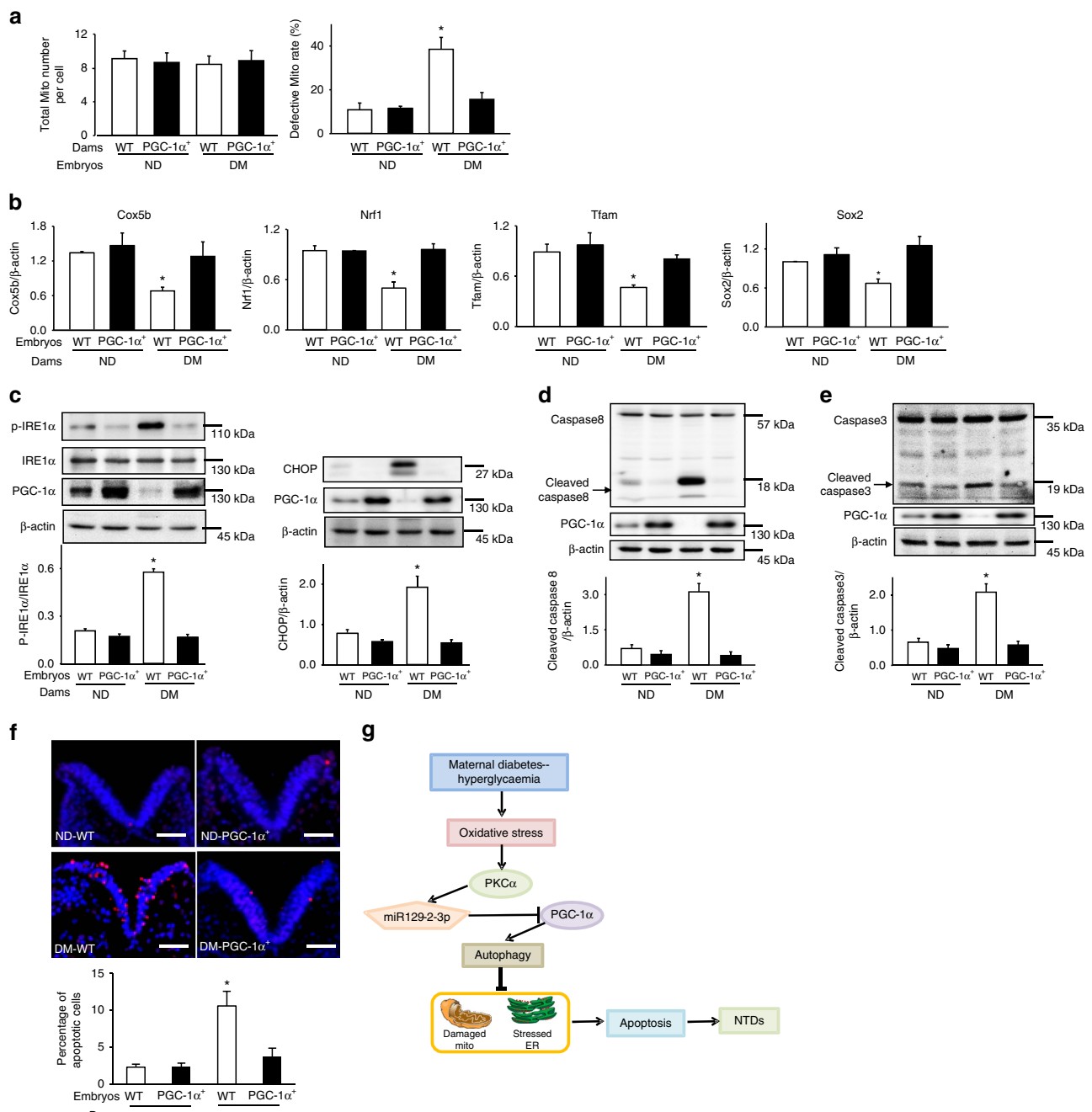

**Figure 9 | PPARGC1A overexpression prevents diabetes-induced cellular organelle stress and apoptosis.** (**a**) Total number of mitochondria (Mito) and percentages of defective mitochondria (number of defective Mito divided by total number of Mito) from wild-type (WT) and PGC-1α overexpressing embryos. PGC-1α transgenic males mated with nondiabetic (ND) and diabetic (DM) females to generate WT and PGC-1α overexpressing embryos. (**b**) mRNA abundance of mitochondrial genes in whole embryos: Cox5b, Nrf1, Tfam, Sox2. Protein abundance of p-IRE1α and CHOP (**c**), cleaved caspase 8 (**d**) and caspase 3 (**e**) in E8.75 embryos. Bar graphs for protein abundance were quantitative data from three independent experiments. Representative images of the TUNEL assays in E8.75 embryos (**f**). Apoptotic cells were labelled in red and nuclei were labelled by DAPI in blue. The dense blue V shape areas are the neural plates. The level of the body axis was at hindbrain level. The bar graph showed the quantification of apoptotic cell number. Experiments were repeated using five embryos ($N = 5$) from different dams and five images were obtained from each embryo. Scale bars are 30 μm. In **a**–**e** experiments were repeated three times using embryos from three different dams ($N = 3$) per group. *indicates significant differences ($P < 0.05$) compared to the other groups in one-way ANOVA followed by Tukey tests. (**g**) is a schematic diagram depicting the maternal diabetes-induced pathway, PKCα-miR-129-2-PGC-1α, in autophagy impairment leading to mitochondrial dysfunction, ER stress, apoptosis and NTD formation. PKCα upregulates miR-129-2, which in turn downregulates PGC-1α. PKCα and miR-129-2 suppress autophagy whereas PGC-1α stimulates autophagy.

that molecular intermediates downstream of PKC may be required for autophagy inhibition induced by PKC. Maternal diabetes-induced autophagy impairment is associated with mitochondrial dysfunction[12]. The master regulator of

mitochondrial biogenesis and function, PGC-1α, is downregulated by maternal diabetes. PGC-1α is implicated as a positive regulator of autophagy[32,33]. Using both *in vivo* and *in vitro* approaches, our studies for the first time unravelled

PGC-1α as a downstream effector of PKCα. Further studies reveal that PKCα represses autophagy through PGC-1α downregulation. Consistent with the findings in the *Prkca* null embryos, restoring PGC-1α in the neuroepithelium prevents autophagy impairment and ameliorates NTD formation under diabetic conditions. Thus, these findings support a critical role of PGC-1α in embryonic neurulation by stimulating autophagy.

While the information regarding transcriptional regulation of PGC-1α is limited, epigenetic mechanisms are implicated in PGC-1α regulation[46]. Because PKCα regulates miRNA expression[34] and our recent miRNA profiling study showed altered miRNA profiling in embryos exposed to maternal diabetes[47], we turned our attention to miRNAs that potentially act downstream of PKCα and repress PGC-1α expression. miR-129-2 is a predicted candidate for the repression of PGC-1α expression. RNA binding assays and functional studies ascertain that PGC-1α is a target gene of miR-129-2, which suppresses PGC-1α expression by degrading its mRNA and inhibiting its protein translation. Maternal diabetes significantly increases miR-129-2 expression. Most importantly, miR-129-2 indeed acts downstream of PKCα. The mechanism underlying PKCα-induced miR-129-2 warrants further investigation. Our study reveals a new pathway, PKCα-miR-129-2-PGC-1α, in mediating the teratogenic effect of maternal diabetes by inhibiting autophagy.

The PGC-1 family of transcription coactivators, including PGC-1α, PGC-1β and PRC, regulate mitochondrial function and cell viability[48]. PGC-1α is abundantly present in the CNS[48]. Although *PPARGC1A* gene deletion does not affect embryonic development[49], possibly due to the compensation of PGC-1β, embryos with ubiquitous deletion of the *Prc* gene manifest peri-implantation lethality[50] as observed in Atg5 null embryos. However, adult PGC-1α null mice do exhibit CNS dysfunction[49]. This evidence suggests a critical role of the PGC-1 family in embryonic neural development, possibly by regulating autophagy. The present study supports the role of PGC-1α in neurulation. PGC-1α overexpression in the neuroepithelium abrogates maternal diabetes-induced autophagy impairment, resolves cellular homeostatic imbalance by preventing mitochondrial dysfunction and ER stress, and ultimately reduces NTD formation. Furthermore, ectopic overexpression of PGC-1α stimulates autophagy. Thus, PGC-1α is an autophagy promoting factor. It is reasoned that PGC-1α-induced autophagy preserves mitochondrial function by removing maternal diabetes-damaged mitochondria. PGC-1α induces mitochondrial biogenesis and function by increasing gene expression that is essential for mitochondrial proliferation, DNA maintenance, oxidative phosphorylation and ROS detoxification[48]. Therefore, it is possible that PGC-1α overexpression restores cellular homeostasis by directly inducing gene expression essential for mitochondrial function. PGC-1α is a co-activator of the peroxisome proliferator-activated receptor gamma (PPARγ). PPARγ agonists, rosiglitazone and pioglitazone, enhance the action of PGC-1α (ref. 51). Sirtuin activators including resveratrol and SIR1720 can decrease PGC-1α acetylation and, thus, increase PGC-1α activity and its downstream target gene[52]. In studies of ours and others[53,54], resveratrol and SIR1720 can ameliorate diabetes-induced NTDs. Future studies may test the preventive effect of PPARγ agonists on diabetic embryopathy.

ER stress and mitochondrial dysfunction are downstream of the PKCα-miR-192-PGC-1α pathway. We have demonstrated that ER stress is indeed a causal factor in diabetes-induced NTDs[13]. Our recent study used the mitochondrial specific superoxide dismutase 2 to inhibit mitochondrial production of reactive oxygen species and mitochondrial dysfunction leading to amelioration of NTD formation in diabetic pregnancy[55].

Thus, ER stress and mitochondrial dysfunction are causally involved in diabetic teratogenesis. All embryos exposed to diabetes exhibit impaired autophagy. A threshold for autophagy impairment may be required for NTD formation. Nevertheless, the level of averaged autophagy activity for all embryos exposed to diabetes is significantly lower than that in all embryos under nondiabetic conditions. Additionally, restoring autophagy activity reduces diabetes-induced NTDs, supporting the causal role of autophagy impairment in diabetic embryopathy. Autophagy gene *Ambra1* deletion leads to massive neuroepithelial cell apoptosis and NTD formation[7]. If neuroepithelial cells in the neural fold fusion points undergo apoptosis, the neural fold would fail to be fused[56]. Our studies have observed excessive cell apoptosis in the developing neuroepithelium and particularly in the neural fold fusion points leading to neurulation failure[11].

In summary, our study reveals a mechanism underlying maternal diabetes-suppressed autophagy in the neuroepithelium leading to NTD formation. We demonstrate that PKCα negatively regulates autophagy, whereas PGC-1α promotes autophagy. Altered autophagy may also contribute to the aetiology of other defects in diabetic pregnancies. Because autophagy is essential for cardiac morphogenesis[57], maternal diabetes-impaired autophagy may contribute to the induction of heart defects. Future studies may aim to reveal the importance of autophagy in other morphogenetic processes that are affected by maternal diabetes.

## Methods

**Mice.** The procedures for animal use were approved by the University of Maryland Baltimore Institutional Animal Care and Use Committee. WT C57BL/6J, SOD1-transgenic (SOD1-Tg) (#002298)[58], and PKCα knockout (KO) mice (#009068)[20] were purchased from the Jackson Laboratory (Stock No. 009068, Bar Harbor, Maine, USA) and backcrossed with the C57BL/6J strain for ten generations. Nestin promoter driven PGC-1α transgenic (PGC-1α-Tg) mice in C57BL/6J background were generated in the Transgenic Core Facility at the University of Maryland Baltimore. The GFP-LC3 strain was created by Dr Noboru Mizushima[59].

**Model of maternal diabetes-induced NTDs.** We[11–13,27] and others[60–62] have used a rodent model of Streptozotocin (STZ)-induced diabetes in research of diabetic embryopathy. Briefly, 8- to 10-week old female mice were intravenously injected daily with 75 mg kg$^{-1}$ STZ in the tail vein over 2 days to induce diabetes. Diabetes was defined as 12-h fasting blood glucose concentrations greater than or equal to 14 mM which usually occurred at 3–5 days after STZ injections. We did not detect any difference in embryonic development between STZ/insulin-treated and non-STZ-treated mice[63], suggesting a lack of residual toxic effect of STZ in our animal model. Insulin pellets (Linshin, Canada) were implanted subcutaneously in diabetic mice to restore euglycaemia (glucose concentrations: 4–6 mM) before mating[13,27]. On day 5.5 of pregnancy (E5.5), insulin pellets were removed to permit frank hyperglycaemia ($\geq$14 mM glucose), so that the developing embryos were exposed to hyperglycaemia during neurulation (E8–10.5). Embryos were collected at E8.75 (14:00 at E8.5) for biochemical and molecular analyses. At E10.5, embryos were examined under a Leica MZ16F stereomicroscope (Bannockburn, IL, USA) to identify NTDs in a blinded fashion.

**Electron microscopy and GFP-LC3 confocal microscopy.** GFP-LC3-Tg mice were used to quantify autophagosome formation *in vivo*[12]. GFP florescent images in embryonic neuroepithelial cells were recorded by confocal microscopy using a laser scanning microscope (LSM 510 META, ZEISS) with a plan-apochromat ×63 Oil numerical aperture 1.4 objective lens, and excitation wave length for GFP (488 nm) and DAPI (405 nm). All pictures in a given figure were taken with the same setting. GFP-LC3 punctate foci with a diameter greater than or equal to 20 pixels in each cell were calculated by the ImageJ software according to the manufacturer's manual. Thus, the images captured the aggregated GFP-LC3 (GFP-LC3 puncta) fluorescent signal that was much stronger than that of individual GFP-LC3 (ref. 59). In neuroepithelial cells of DM embryos, individual GFP-LC3 protein was diffused in cytoplasm, did not form puncta and had a much lower fluorescent signal that was not captured in the images.

Mitochondrial structures were examined by transmission electron microscopy in our university's electron microscopy core facility. Thick sections (1 μm) were cut and visualized at ×100 magnification to identify the neuroepithelia of the E8.75 embryos. Thin sections (80 nm) of identified neuroepithelia were cut and viewed

with an electron microscope (Joel JEM-1200EX; Tokyo, Japan) at high resolution (10, 12 and 25 K) to identify the cellular organelle structures.

**TUNEL assay.** ApopTag Red *In Situ* Apoptosis Detection Kit (Catalog No: S7165, Millipore) was used to detect apoptosis[11]. Ten-μm frozen embryonic sections were fixed with 4% PFA in PBS and incubated with TUNEL reaction agents. The percentage of apoptotic cells was obtained by dividing the number of TUNEL-positive cells with the total number of cells in a microscopic field and then multiplying by 100 from three separate experiments.

**Immunoblotting.** Protein (30–50 μg) from one embryo was used and embryos were collected from different dams. Embryos were lysed in a lysis buffer (Cell Signaling, #9803) with a protease inhibitor cocktail (Sigma). Mitochondrial proteins were extracted by the Mitochondria Isolation Kit from Thermo Scientific (#89801). Immunobilon-P or Immunobilon-P$^{SQ}$ (Milllipore) membranes were used for immunoblotting. Membranes were exposed to goat anti-rabbit or goat anti-mouse (Jackson ImmunoResearch Laboratories) or goat anti-rat (Chemicon) secondary antibodies. Signals were detected using SuperSignal West Femto Maximum Sensitivity Substrate kit (Thermo Scientific) and chemiluminescence emitted from the bands was directly captured using the UVP Bioimage EC3 system. Densitometric analyses of chemiluminescence signals were performed using the VisionWorks LS software (UVP). All uncropped western blots can be found in Supplementary Fig. 7.

**Transfection and mitochondrion/autophagosome imaging.** C17.2 mouse neural stem cells, originally obtained from ECACC (European Collection of Cell Culture), are newborn mouse cerebellar progenitor cells transformed with retroviral v-myc[21,22]. HeLa cells stably expressing LC3-GFP[40] were kindly provided by Dr Shengyun Fang in the University of Maryland Baltimore. C17.2 cells were transfected with the scramble control siRNA, the PKCα-siRNA (sc-35960, Santa Cruz Biotechnology), the PGC1α-siRNA (sc-38885) or the control siRNA using Lipofectamine RNAiMAX (Invitrogen) according to the manufacture's protocol. There were no mycoplasma contamination in the C17.2 cell line and the LC3-GFP HeLa cell line.

After seeded for 24 h, LC3-GFP Hela cells were transfected with the PGC-1α vector (Addgene) using Lipofectamine 2000. Cells were collected at different time points with additions of MitoID (Enzo life sciences, Farmingdale, NY, USA) for mitochondrial staining and Cyto-ID (the Cyto-ID autophagy detection kit, Cat# ENZ-51031-0050, Enzo life sciences). The Cyto-ID green dye is specific for selectively staining autophagic vesicles in living cells. According to the manufacturer's instructions, yhe Cyto-ID autophagy staining, which detects pre-autophagosomes, autophagosomes and autolysosomes, was validated with known inhibitors and activators of autophagic activity. Briefly, C17.2 cells were incubated in $1\times$ Assay Buffer added with Cyto-ID green dye for 30 min at 37 °C and protected from light. After washes with $1\times$ Assay Buffer, cells were fixed with 4% PFA in PBS, washed with $1\times$ Assay Buffer and covered with cover slides. Then the stained cells were analysed by the Nikon Ni-U microscope with a Plan Apo $\times20$ numerical aperture 0.75 objective lens and the Iplab software (Qimaging, V3.95). All pictures in a given figure were taken with the same setting.

**Biotin-labelled miR-129-2 pulldown assay.** The biotin-labelled miR-129-2-3p (Dharmacon, Lafayette, CO, USA) was transfected into C17.2 cells for 48 h, and then whole-cell lysates were collected. Cell lysates were mixed with streptavidin-coupled Dynabeads (Invitrogen) and incubated at 4 °C on rotator overnight. After the beads were washed thoroughly, the bead-bound RNA was isolated and subjected to RT followed by real-time PCR analysis. Input RNA was extracted and served as a control.

**Determination of the miR-129-2 binding site on PGC-1α mRNA.** The full-length PGC1α coding region (CR) or its 3′-UTR and 3′-UTR fragments with the predicted miR-129-2 binding site (BS) or mutated-miR-129-2 binding site (BS-Mut) were amplified and subcloned into the pmirGLO Dual-Luciferase miRNA Target Expression Vector (Promega, Madison, WI, USA) to generate the pmirGLO-Luc-PGC1α-CR and pmirGLO-PGC1α-3′UTR and pmirGLO-PGC1α-BS and pmir-GLO-PGC1α-BS-Mut. Luciferase activities were measured using the Dual-Luciferase Assay System (Promega), and were normalized by the *Renilla* luciferase activity. Real-time PCR (RT–PCR) and subsequent calculations were performed by the StepOnePlus Real-Time PCR System (Applied Biosystem). All primer sequences were listed in Supplementary Table 3.

**Statistical analysis.** Sample size was estimated to achieve 80% power based on our previous study[11]. Nondiabetic and diabetic dams were randomly assigned to different experimental groups. Statistical differences were determined by Student's *t*-test for two group comparisons and one-way ANOVA for more than two group comparisons using the SigmaStat 3.5 software. In ANOVA analyses, Tukey tests were used to estimate the significance of the results. Significant difference between groups in NTD incidences was analysed by the chi-square test. The variance was similar between the groups that were being statistically compared.

**Data availability.** The authors declare that all data supporting the findings of this study are available within the article and its supplementary information files or from the corresponding author on reasonable request.

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

## Acknowledgements

We thank Ms Hua Li for her technical support and Dr Julie Wu at the University of Maryland Baltimore for her assistance in preparing this manuscript. This work was supported by the NIH NIDDK grants R01DK083243, R01DK101972, R01HL131737 (P.Y.) and R01DK103024 (to P.Y. and E.A.R.), and an American Diabetes Association Basic Science Award (1-13-BS-220) (P.Y.). We thank the support from the Office of Dietary Supplements, National Institute of Health (NIH).

## Author contributions

F.W., C.X., Y.W., X.L., J.Y., D.D., Pe.Y. and J.Z. researched the data. P.Y. conceived the project, designed the experiments and wrote the manuscript. E.A.R., C.W. and C.H. participated in data analyses and reviewed the manuscript.

## Additional information

**Competing interests:** The authors declare no competing financial interests.

