## [Peer Review File · Nature Communications]

Reviewer #1 (Remarks to the Author)

The authors have studied the process by which maternal diabetes induces neural tube defects (NTDs), and suggest the existence of a novel pathway;

PKCa---miR-129-2---PGC-1a

which inhibits autophagy in the developing neuroepithelium.

Their study provides a mechanistic basis for targeting PKC α and miR-129-2, and suggests that the use of PGC-1 α agonists may prevent NTDs in diabetic pregnancy.

The authors report that key autophagy regulators modulates by maternal diabetes in the developing neuroepithelium. For instance, maternal diabetes causes exencephaly and induces neuroepithelial cell apoptosis and suppresses autophagy in the forebrain and midbrain of NTD embryos. When the authors deleted the *Prkca* gene, which encodes for PKCa, they were able to reverse the diabetes-induced autophagy impairment, as well as the diabetes-induced cellular organelle stress and apoptosis, which resulted in a reduction in the NTD incidence. Furthermore, they report that PKCa increases the expression of miR-129-2, which turns out to be a negative regulator of autophagy. Thus, miR-129-2 diminish autophagy by directly inhibiting the master metabolic regulator, PGC-1a, which supports neurulation by acting as an autophagy activator in neuroepithelial cells.

These novel findings identified two negative regulators of autophagy, PKCa and miR-129-2, which mediate the teratogenicity of hyperglycemia leading to NTD formation. The authors also revealed a new function for PGC-1a in embryonic development through promoting autophagy and ameliorating hyperglycemia-induced NTDs.

This is a rich paper with an impressive amount of results. Still, it is easy to read and comprehend, due to a logical organization of the data and a clear and precise language.

I find the argumentation in favor of the suggested pathway inspiring and convincing.

My comments concern the ramifications of the presented work.

1. The authors identify PKCa activation as the primary culprit of the NTD induction, and prove their point by constructing a mouse PKCa-KO strain, in which the diabetes-induced NTDs are largely blocked. However, there are reports of increased diabetes-induced activity of other PKC enzyme than PKCa, for instance PKC β I, β II, and γ . In the view of the authors, how much teratological "cross-talk" would they expect between the classic PKC isoforms? Have they measured – and tried to affect – the activities of other isoforms of PKC?

2. Another line of evidence is the overexpression of PGC-1a in transgenic mice, which, consequently leads to less NTDs in the offspring of diabetic animals compared to the offspring of diabetic WT mice. This is a novel approach, which may have therapeutical implications, given that there are PGC-1a agonists that are harmless to the embryo (and mother). What types of agonists do the authors know about, and have they tried any of them in their experimental system?

3. The authors also find that PKCa KO leads to normalization of several other negative effects of the diabetic environment – such as ER stress and mitochondrial dysfunction. Are these processes, which have been indicated by other studies, also causally involved in diabetic teratogenesis, or are they rather side phenomenon?

4. The authors allow a diabetic period in pregnancy between embryo (E) days 5.5 – 10.5, by

removing insulin pellets from the neck of the pregnant mice. What would they expect to find, if they allowed more time to pass, and studied the outcomes at later gestational stages, say days 12.5, 15.5 and 18.5?

5. The analogous question would be if they would allow diabetes in their animals the whole gestation (without inserting insulin pellets before pregnancy) – would they then find other types of malformations with other types of severity?

6. Have the authors tried to test their findings in a non-mouse model, e.g. in a rat or rabbit model?

Reviewer #2 (Remarks to the Author)

The manuscript by Wang et al investigates the molecular and cellular changes responsible for the increase in neural tube defects (NTDs) in embryos associated with diabetic pregnancies. In humans, maternal diabetes is associated with an increase in embryopathies, including NTDs. Similar birth defects can be phenocopied in mouse models using the NOD mouse strain or Streptozotocin treatment. Using the Streptozotocin model, Wang et al examine the proposed link between diabetes and autophagy as it relates to NTDs. The findings, using genetically modified mice as well as gene knockdown and gene over expression studies in vitro, support the general conclusions of their model in which PKC alpha increases miRNA expression, and this in turn inhibits PGC-1 alpha induced autophagy which then causes NTDs. The study is original and should be of general interest as it provides a potential mechanism for the increase in NTDs seen in maternal diabetes for this mouse model as well as possible ways to suppress this pathology. Nevertheless, there are several issues with manuscript that reduce enthusiasm at this time.

Main concerns:

1. NTDs can be caused by defects in several tissues and are not limited to autonomous defects in the neuroepithelium. Although some of the studies presented focus on the neuroepithelium – examination of apoptosis, autophagy and stress for example – many others use whole embryos for analysis, such as those studying protein levels in Figures 6 and 7. The manuscript needs to distinguish very carefully which results were found in the neuroepithelium and which were only studied in the context of the whole embryo. The experiments in Figure 9 using the nestin promoter are supportive of a direct effect of PGC-1alpha on the neuroepithelium, but those using the Prkca-null mouse in earlier figures cannot distinguish between systemic and local effects related to the formation of NTDs.

2. The manuscript needs to address other recent publications that have proposed potentially different mechanisms for the origin of NTDs in association with maternal diabetes. For example the paper of Salbaum et al, Scientific Reports 5, 16917, (2015), proposed a defect in mesoderm formation and the primitive streak in such pathology. It is possible that these two studies could be reconciled if Prkca was acting on the mesoderm rather than the neuroepithelium.

3. Similarly, the authors should address the partial penetrance of the NTD phenotype even though they report consistent changes in autophagy, protein levels, mRNA levels etc in multiple wild-type embryos from diabetic dams.

4. It would be of interest to determine if loss of PGC 1alpha exacerbated the incidence of NTDs in embryos associated with diabetic mothers as a further test of the hypothesis, but perhaps this is for the future if the data are not currently available.

Other issues:

1. The logic of the arguments presented in the Introduction is not always clear. For example,

"However, it is unclear how maternal diabetes represses autophagy during neurulation", but at this point there has been no mention of a connection between diabetes and autophagy.

2. Figure 1 could be amended to a Table of NTD incidence, and the other data could be moved to a supplementary figure. Also with respect to this figure and the sections shown, throughout the manuscript the authors fail to discuss whether the embryos sectioned have a consistent mutant phenotype, for example, exencephaly versus spina bifida or microcephaly.

3. For Figure 2B, several of the sections are shown magnified in adjacent panels, and yet in the legend it states that the scale bars are 300um in all panels. This is not correct.

4. Figure 2C requires labels or a key to indicate the meaning of each of the three bars.

5. Some figures label the mouse model as Prkca and others as Pkca (e.g. Fig 5). The nomenclature should be standardized.

6. Figure 5F and legend. "** indicate significant difference with other group or groups" This statement is ambiguous and it is not clear if the 25mM glucose control siRNA sample is the only one significantly different from all the others.

7. Figure 9E. It is not clear why the 30 and 42 hour time points for LC3-GFP are not also significantly raised compared to the early time points as stated in the legend, or do the authors mean the previous time point?

8. Figure S3. Panel A. The legend states that the abundance of U6 RNA is shown, but this is not so. Perhaps they axis is mislabeled for the first bar graph shown, and also "bounds" should be bound.

9. Figure S3. Panel F. Based on the hypothesis presented, the expectation would be that increased levels of an miR129 inhibitor would enable increased PGC 1alpha expression compared to a control inhibitor, but this does not occur. The authors should explain this result in more detail.

10. Figure S3. The text referring to panel G needs to be indicated in the legend.

11. Figure S4 legend. Autophagesome = autophagosome (two instances) and Staining punctate = staining puncta OR punctate staining (two instances).

12. Figure S5 legend does not relate to the Figure shown. Part of it is derived from Figure 10 legend.

Reviewer #3 (Remarks to the Author)

This paper investigates the molecular mechanism downstream of high glucose that is responsible for neural tube closure defects (NTDs) in mouse embryos of diabetic mothers. The frequency of many birth defects is higher in diabetic mothers and so understanding the mechanisms of this effect is of considerable importance. The authors follow up previous work showing that autophagy is required for neural tube closure, and is diminished in embryos of diabetic mothers. They identify a sequence of molecular events, in which hyperglycemia induces PKCalpha, which increases the expression of miR-129-2, a negative regulator of autophagy via inhibition of PGC-1alpha. This is a significant step forward in understanding the molecular mechanism by which diabetes confers an increased risk of NTDs.

Particular strengths of the study are the use of PKCalpha null and PGC-1alpha over-expressing mice (the latter made for this study), that enable very clear dissection of the pathway. I am also impressed that the authors have analysed NTD frequency at E10.5, when neural tube closure should have been completed, and then have carefully gone to the earlier E8.75 stage to analyse

neuroepithelial events involved in the pathogenesis. In this way they have avoided the common mistake of using post-neurulation embryo stages to seek mechanisms of failed neurulation, when secondary effects of failed closure may be detected. Cultured cell studies have been used to reveal specific cell biological aspects of the pathway, but the authors have frequently returned to the in vivo embryonic system to validate findings, which is an attractive aspect of the study.

One issue that deserves comment (in the Discussion of a revised paper) is why autophagy is required, and apoptosis detrimental, to neural tube closure. The authors just state that lack of autophagy and induction of apoptosis results in NTDs, but they do not explain this link. They say that: "... apoptosis has to reach a threshold for NTD formation ..." (p7), but why could a smaller neuroepithelium (that has lost cells due to apoptosis) not be still capable of closing to form a neural tube? Are other mechanisms involved? This point should be discussed in relation to the neurulation literature.

The data are clear and, for the most part well presented. However, I have a number of suggestions for improvement of the figures, to enhance readability, as follows:

Fig 1. Please state clearly in the legend which type of NTD (exencephaly?) is shown in the embryo in (B) and sections in (C).

Fig 2. The quantitation of apoptosis requires normalisation to total cell (nuclear) number in the sections that are being compared. If total cell number was greater per section in the DM-NTD than DM or ND, then an increased apoptotic count would be expected, even if there was no difference in apoptosis incidence. The authors should either present data on average nuclear number per section for the three treatments (do they differ?), or actually calculate apoptotic indices (no. apoptotic nuclei/total no. nuclei). In addition, the graphs in (C) require a key to define the bar colors, or insert labels on the X-axis.

Fig 3. The "green LC3-GFP puncta" which are taken as evidence of autophagic activity are not visible in these low magnification images of whole embryo sections. Please insert some specimen high magnification images to show the puncta clearly, or refer to Fig 5 where they are shown.

Fig 4. Although the statistical comparison data for (C) are shown separately in Suppl Table 1, the statistically significant differences should also be indicated on the graphs, to make the result easier for the reader to appreciate. Also applies to Fig 9H.

Fig 5. It would aid the reader if the statistical differences were shown more clearly on graphs. Placing an asterisk over one bar does not indicate which other bar(s) it was compared with. Please insert tie-lines to link bars that were compared statistically, with overlying asterisks. Same comment applies to Figs 2C and Figs 6-10.

Fig 6. Please refer to the images of defective and normal mitochondria in Suppl Fig 1, to illustrate how the distinction was made for quantitation in this figure. In Suppl Fig 1, please indicate the normal and defective mitochondria by labels or arrows.

Fig 7. "The dense blue V shape areas are the neural tubes." This description is not accurate, as at E8.75 this is neural plate that has not yet closed to form the neural tube. Please re-phrase and also indicate which level of the body axis (hindbrain?) the images are showing. Also applies to Figs 9A and 10F.

Fig 8, D-F. Did the authors consider looking at cells exposed to caPKCalpha or the miR-129-2 mimic in combination with high glucose? Is there an additive effect? Please label the Y-axis in (A) to clearly indicate that this is relative mRNA level, as already done in (F) and (H).

Fig 9. The lettering on the construct in (A) is too small to read. Please enlarge.

Reviewer #4 (Remarks to the Author)

Comments on Wang et al

The paper by Wang et al. investigates the mechanisms of diabetes induced neuronal tube defects. In particular, the authors show that diabetes activates PKC α , which in turn increases the expression of micro-RNA 129-2. This micro-RNA subsequently represses the expression of PGC-1 α . Since PGC-1 α is an activator of autophagy, this pathway is shown here to inhibit autophagy. The authors further show that overexpression of PGC-1 α in the neuroepithelium during development ameliorates the NTD defects caused by diabetes. The manuscript therefore reports a strong correlation between PKC α activation, PGC-1 α expression and autophagy, although it is not clear to which extent all beneficiary effects of PGC-1 α expression are due to autophagy induction.

In summary, the authors report a detailed and mechanistic study that will be interested for scientists working on diabetes, NTDs and autophagy.

There are a number of comments and questions the authors should address:

Major points:

1. From Fig. 5B and 9D it is unclear which structures were considered as puncta for the quantifications shown to the right. It seems that there is generally less GFP-LC3 in DM-WT cells. The blots shown in Fig. 5C and to some extent also Fig. 9C also suggest that there is less LC3 in DM-WT cells. While the LC3 in Fig. 5C and 9C are the endogenous proteins and could be transcriptionally regulated, I understand that GFP-LC3 is a transgene and should therefore not be subject to the same regulation. Do the authors have an explanation for this?
2. Related to the point above, the authors quantify the total amount of LC3-II in Fig. 5C and it seems clear that there is less LC3-II in DM-WT cells compared to Prkca $^{-/-}$ DM cells. However, the total amount of LC3 is also lower in DM-WT cells. Can the authors provide a value for the ration of LC3-II/LC3-I?
3. Can the authors explain how cyto-ID staining works (Fig. 5D-F)? This information is important to interpret the results.
4. In Fig. 5F the authors count over 500 puncta and by implication autophagosomes/cell. This is a very high number, in particular as the quantification of E8.75 neuroepithelial cells yields values that are about 50-100 times lower (Fig. 5B). Are authors sure that their quantification methods count individual autophagosomes?
5. The colocalization of GFP-LC3 with MitoID shown in Fig. S4B is not very convincing as the signals are very diffuse. Either the authors show better pictures or they tone down their statements.
6. The data for Fig. 3B should be quantified
7. Fig. 4A, B should be quantified.
8. Details about the fluorescent microscopy setting are missing. Were all pictures in a given Figure taken with the same settings?
9. For Fig. 5A and 9B the authors should indicate how many autophagosomes were counted in total.

10. The authors write that 83% of NTDs were comprised of exencephaly (Page 7). How do the authors get to this number?

Minor points

11. The x-axis in Fig. 2C needs labels

12. The second sentence of the "In Brief" section is unclear.

Responses to Reviewers' Concerns

Reviewers' comments:

Reviewer #1 (Remarks to the Author):

The authors have studied the process by which maternal diabetes induces neural tube defects (NTDs), and suggest the existence of a novel pathway; PKC α ---miR-129-2---PGC-1 α , which inhibits autophagy in the developing neuroepithelium. Their study provides a mechanistic basis for targeting PKC α and miR-129-2, and suggests that the use of PGC-1 α agonists may prevent NTDs in diabetic pregnancy. The authors report that key autophagy regulators modulates by maternal diabetes in the developing neuroepithelium. For instance, maternal diabetes causes exencephaly and induces neuroepithelial cell apoptosis and suppresses autophagy in the forebrain and midbrain of NTD embryos. When the authors deleted the *Prkca* gene, which encodes for PKC α , they were able to reverse the diabetes-induced autophagy impairment, as well as the diabetes-induced cellular organelle stress and apoptosis, which resulted in a reduction in the NTD incidence. Furthermore, they report that PKC α increases the expression of miR-129-2, which turns out to be a negative regulator of autophagy. Thus, miR-129-2 diminish autophagy by directly inhibiting the master metabolic regulator, PGC-1 α , which supports neurulation by acting as an autophagy activator in neuroepithelial cells.

These novel findings identified two negative regulators of autophagy, PKC α and miR-129-2, which mediate the teratogenicity of hyperglycemia leading to NTD formation. The authors also revealed a new function for PGC-1 α in embryonic development through promoting autophagy and ameliorating hyperglycemia-induced NTDs.

This is a rich paper with an impressive amount of results. Still, it is easy to read and comprehend, due to a logical organization of the data and a clear and precise language.

I find the argumentation in favor of the suggested pathway inspiring and convincing.

Response: We greatly appreciate the reviewer's positive comments.

My comments concern the ramifications of the presented work.

1. The authors identify PKC α activation as the primary culprit of the NTD induction, and prove their point by constructing a mouse PKC α -KO strain, in which the diabetes-induced NTDs are largely blocked. However, there are reports of increased

diabetes-induced activity of other PKC enzyme than PKC α , for instance PKC β I, β II, and γ . In the view of the authors, how much teratological "cross-talk" would they expect between the classic PKC isoforms? Have they measured – and tried to affect – the activities of other isoforms of PKC?

Response: Our previous studies have demonstrated that besides PKC α , maternal diabetes also activates PKC β II and PKC δ (Am J Obstet Gynecol. 2011 Jul;205(1):84.e1-6; Reprod Sci. 2008 Apr;15(4):349-56). Specific inhibitors of PKC β II and PKC δ reduced high glucose in vitro-induced NTDs (Am J Obstet Gynecol. 2011 Mar;204(3):226.e1-5; Reprod Sci. 2008 Apr;15(4):349-56). These evidence suggest that these three PKC isoforms have distinctive roles in the etiology of diabetic embryopathy.

We include the above statement in the Paragraph 2 of the Discussion.

2. Another line of evidence is the overexpression of PGC-1a in transgenic mice, which, consequently leads to less NTDs in the offspring of diabetic animals compared to the offspring of diabetic WT mice. This is a novel approach, which may have therapeutical implications, given that there are PGC-1a agonists that are harmless to the embryo (and mother). What types of agonists do the authors know about, and have they tried any of them in their experimental system?

Response: We appreciate the reviewer's constructive comment. PGC-1 α is a co-activator of the peroxisome proliferator-activated receptor gamma (PPAR γ). PPAR γ agonists, rosiglitazone and pioglitazone, can enhance the action of PGC-1 α (Neurochem Res, 2015, 40:308-316). Sirtuin activators including resveratrol and SIR1720 can decrease PGC-1 α acetylation and, thus, increase PGC-1 α activity and its downstream target gene (Cell, 2006, 127:1109-1122). In studies of ours and others (J Neurochem. 2016 May;137(3):371-83; Mol Nutr Food Res. 2011 Aug;55(8):1186-96; Reprod Sci. 2012 Sep;19(9):949-61.), resveratrol and SIR1720 can ameliorate diabetes-induced NTDs. Future studies may test the preventive effect of PPAR γ agonists on diabetic embryopathy.

We include the above statement in the Paragraph 5 of the Discussion.

3. The authors also find that PKC α KO leads to normalization of several other negative effects of the diabetic environment – such as ER stress and mitochondrial dysfunction. Are these processes, which have been indicated by other studies, also causally involved in diabetic teratogenesis, or are they rather side phenomenon?

Response: ER stress and mitochondrial dysfunction are downstream of the PKC α -miR-192-PGC-1a pathway. We have demonstrated that ER stress is indeed a causal factor in diabetes-induced NTDs (Diabetes. 2013 Feb;62(2):599-608.). Our recent study used the mitochondrial specific superoxide dismutase 2 to inhibit mitochondrial production of reactive oxygen species and mitochondrial dysfunction

leading to amelioration of NTD formation in diabetic pregnancy (*Free Radic Biol Med.* 2016 Jul;96:234-44.). Thus, ER stress and mitochondrial dysfunction are causally involved in diabetic teratogenesis.

We include the above statement in the Paragraph 6 of the Discussion.

4. The authors allow a diabetic period in pregnancy between embryo (E) days 5.5 – 10.5, by removing insulin pellets from the neck of the pregnant mice. What would they expect to find, if they allowed more time to pass, and studied the outcomes at later gestational stages, say days 12.5, 15.5 and 18.5?

Response: At later gestational stages, we found congenital heart defects associated or not associated with NTDs (*Circ Cardiovasc Genet.* 2015 Oct;8(5):665-76). NTD embryos will die before birth.

5. The analogous question would be if they would allow diabetes in their animals the whole gestation (without inserting insulin pellets before pregnancy) – would they then find other types of malformations with other types of severity?

Response: Our recent study did not use insulin during early gestation and had the same types and a similar rate of NTD (*Free Radic Biol Med.* 2016 Jul;96:234-44.). Thus, insulin treatment during pre-implantation period did not affect the severity and rate of birth defects.

6. Have the authors tried to test their findings in a non-mouse model, e.g. in a rat or rabbit model?

Response: Our previous studies used the rat as the animal model (*Am J Obstet Gynecol.* 2006 Feb;194(2):580-5). In order to use genetically modified mice in uncovering mechanistic insights, we switched from rats to mice a decade ago.

--

Reviewer #2 (Remarks to the Author):

The manuscript by Wang et al investigates the molecular and cellular changes responsible for the increase in neural tube defects (NTDs) in embryos associated with diabetic pregnancies. In humans, maternal diabetes is associated with an increase in embryopathies, including NTDs. Similar birth defects can be phenocopied in mouse models using the NOD mouse strain or Streptozotocin treatment. Using the Streptozotocin model, Wang et al examine the proposed link between diabetes and autophagy as it relates to NTDs. The findings, using genetically modified mice as well as gene knockdown and gene over expression studies in vitro, support the general conclusions of their model in which PKC alpha

increases miRNA expression, and this in turn inhibits PGC-1 alpha induced autophagy which then causes NTDs. The study is original and should be of general interest as it provides a potential mechanism for the increase in NTDs seen in maternal diabetes for this mouse model as well as possible ways to suppress this pathology. Nevertheless, there are several issues with manuscript that reduce enthusiasm at this time.

Main concerns:

1. NTDs can be caused by defects in several tissues and are not limited to autonomous defects in the neuroepithelium. Although some of the studies presented focus on the neuroepithelium – examination of apoptosis, autophagy and stress for example – many others use whole embryos for analysis, such as those studying protein levels in Figures 6 and 7. The manuscript needs to distinguish very carefully which results were found in the neuroepithelium and which were only studied in the context of the whole embryo. The experiments in Figure 9 using the nestin promoter are supportive of a direct effect of PGC-1alpha on the neuroepithelium, but those using the Prkca-null mouse in earlier figures cannot distinguish between systemic and local effects related to the formation of NTDs.

Response: We have distinguished very carefully for results from the neuroepithelium or the whole embryo in the figure legend and the text. We agree that using the Prkca-null mouse cannot distinguish between systemic and local effects related to NTD formation, and made it clear that the results from the Prkca-null mouse model are from the whole embryo level.

2. The manuscript needs to address other recent publications that have proposed potentially different mechanisms for the origin of NTDs in association with maternal diabetes. For example the paper of Salbaum et al, Scientific Reports 5, 16917, (2015), proposed a defect in mesoderm formation and the primitive streak in such pathology. It is possible that these two studies could be reconciled if Prkca was acting on the mesoderm rather than the neuroepithelium.

Response: We have discussed this issue by citing this paper in Paragraph 2 of the Discussion.

A recent study suggests that maternal diabetes-induced defects in mesoderm formation and the primitive streak cause NTD formation in later stages. Future studies may aim to reveal whether PKC α is activated in the mesoderm and the primitive streak and determine whether deleting the Prkca gene specifically in mesoderm lineage reduces diabetes-induced structural birth defects.

3. Similarly, the authors should address the partial penetrance of the NTD phenotype even though they report consistent changes in autophagy, protein levels, mRNA levels etc in multiple wild-type embryos from diabetic dams.

Response: We observed that all embryos exposed to diabetes exhibit impaired autophagy. A threshold for autophagy impairment may be required for NTD formation. Nevertheless, the level of averaged autophagy activity for all embryos exposed to diabetes is significantly lower than that in all embryos under nondiabetic conditions. Additionally, restoring autophagy activity reduces diabetes-induced NTDs, supporting the causal role of autophagy impairment in diabetic embryopathy.

We include the above statement in the Paragraph 6 of the Discussion.

4. It would be of interest to determine if loss of PGC 1alpha exacerbated the incidence of NTDs in embryos associated with diabetic mothers as a further test of the hypothesis, but perhaps this is for the future if the data are not currently available.

Response: We greatly appreciate the reviewers' constructive comments. We have obtained the PGC-1 α Floxed mice and will examine if deleting the PGC-1 α gene in the neuroepithelium will enhance NTD formation in diabetic pregnancy.

Other issues:

1. The logic of the arguments presented in the Introduction is not always clear. For example, "However, it is unclear how maternal diabetes represses autophagy during neurulation", but at this point there has been no mention of a connection between diabetes and autophagy.

Response: We have introduced that maternal diabetes suppresses autophagy in the developing neuroepithelium before this sentence.

2. Figure 1 could be amended to a Table of NTD incidence, and the other data could be moved to a supplementary figure. Also with respect to this figure and the sections shown, throughout the manuscript the authors fail to discuss whether the embryos sectioned have a consistent mutant phenotype, for example, exencephaly versus spina bifida or microcephaly.

Response: We moved Figure 1 to the supplementary Figure 1, and indicated embryonic NTD phenotypes in sectioned embryos.

3. For Figure 2B, several of the sections are shown magnified in adjacent panels, and yet in the legend it states that the scale bars are 300um in all panels. This is not correct.

Response: Different scale bars are indicated in the figure legend.

4. Figure 2C requires labels or a key to indicate the meaning of each of the three bars.

Response: The bars have been labeled.

5. Some figures label the mouse model as Prkca and others as Pkca (e.g. Fig 5). The nomenclature should be standardized.

Response: We have standardized the nomenclature as Prkca.

6. Figure 5F and legend. "* indicate significant difference with other group or groups" This statement is ambiguous and it is not clear if the 25mM glucose control siRNA sample is the only one significantly different from all the others.

Response: The 25 mM glucose control siRNA group is significantly different from all the others, and the 5 mM glucose control siRNA group is also significantly different from all the others. We indicate significant difference when compared with the 25 mM glucose control siRNA group.

7. Figure 9E. It is not clear why the 30 and 42 hour time points for LC3-GFP are not also significantly raised compared to the early time points as stated in the legend, or do the authors mean the previous time point?

Response: Yes, it means the previous time point. A correction has been made in the legend of Figure 9E.

8. Figure S3. Panel A. The legend states that the abundance of U6 RNA is shown, but this is not so. Perhaps they axis is mislabeled for the first bar graph shown, and also "bounds" should be bound.

Response: It should be the abundance of PGC-1 α mRNA. Corrections have been made for the mislabeling and "bounds" to "bound".

9. Figure S3. Panel F. Based on the hypothesis presented, the expectation would be

that increased levels of an miR129 inhibitor would enable increased PGC 1alpha expression compared to a control inhibitor, but this does not occur. The authors should explain this result in more detail.

Response: We have explained in more detail. The purpose of Fig. 3F and 3E was to determine the optimal dose of the miR-129-2 inhibitor, which effectively reduced miR-129-2 but did not affect the expression of PGC-1 α . Under normal glucose condition, PGC-1 α expression reached a plateau, which cannot be further increased by the miR-129-2 inhibitor. Thus, 50 nM miR-129-2 inhibitor slightly increased PGC-1 α expression, whereas high concentrations of the miR-129-2 inhibitor reduced PGC-1 α expression, probably due to cell toxicity. On the other hand, 50 nM miR-129-2 inhibitor blocked high glucose-repressed PGC-1 α expression (Fig. 7G, H, now Fig. 6G, H)

We have included this clarification in the Results.

10. Figure S3. The text referring to panel G needs to be indicated in the legend.

Response: Figure S3G (Now Fig. S4G) is indicated in the legend.

11. Figure S4 legend. Autophagesome = autophagosome (two instances) and Staining punctate = staining puncta OR punctate staining (two instances).

Response: They were corrected.

12. Figure S5 legend does not relate to the Figure shown. Part of it is derived from Figure 10 legend.

Response: Thank you for pointing out this mistake. We corrected the legend.

Reviewer #3 (Remarks to the Author):

This paper investigates the molecular mechanism downstream of high glucose that is responsible for neural tube closure defects (NTDs) in mouse embryos of diabetic mothers. The frequency of many birth defects is higher in diabetic mothers and so understanding the mechanisms of this effect is of considerable importance. The authors follow up previous work showing that autophagy is required for neural tube closure, and is diminished in embryos of diabetic mothers. They identify a sequence

of molecular events, in which hyperglycemia induces PKC α , which increases the expression of miR-129-2, a negative regulator of autophagy via inhibition of PGC-1 α . This is a significant step forward in understanding the molecular mechanism by which diabetes confers an increased risk of NTDs.

Particular strengths of the study are the use of PKC α null and PGC-1 α over-expressing mice (the latter made for this study), that enable very clear dissection of the pathway. I am also impressed that the authors have analysed NTD frequency at E10.5, when neural tube closure should have been completed, and then have carefully gone to the earlier E8.75 stage to analyse neuroepithelial events involved in the pathogenesis. In this way they have avoided the common mistake of using post-neurulation embryo stages to seek mechanisms of failed neurulation, when secondary effects of failed closure may be detected. Cultured cell studies have been used to reveal specific cell biological aspects of the pathway, but the authors have frequently returned to the in vivo embryonic system to validate findings, which is an attractive aspect of the study.

One issue that deserves comment (in the Discussion of a revised paper) is why autophagy is required, and apoptosis detrimental, to neural tube closure. The authors just state that lack of autophagy and induction of apoptosis results in NTDs, but they do not explain this link. They say that: "... apoptosis has to reach a threshold for NTD formation ..." (p7), but why could a smaller neuroepithelium (that has lost cells due to apoptosis) not be still capable of closing to form a neural tube? Are other mechanisms involved? This point should be discussed in relation to the neurulation literature.

Response: Autophagy gene Ambra1 deletion leads to massive neuroepithelial cell apoptosis and NTD formation (*Nature* **447**, 1121-1125, 2007). If neuroepithelial cells in the neural fold fusion points undergo apoptosis, the neural fold would fail to be fused (Epithelial fusion during neural tube morphogenesis, *Birth Defects Res A Clin Mol Teratol.* 2012 Oct; 94(10): 817–823.). Our studies have observed excessive cell apoptosis in the developing neuroepithelium and particularly in the neural fold fusion points leading to neurulation failure (*Science signaling* 6, ra74, doi:10.1126/scisignal.2004020 (2013).).

The above statement is included in Paragraph 6 of the Discussion.

The data are clear and, for the most part well presented. However, I have a number of suggestions for improvement of the figures, to enhance readability, as follows:

Fig 1. Please state clearly in the legend which type of NTD (exencephaly?) is shown in the embryo in (B) and sections in (C).

Response: We have stated clearly in the figure legend (now is supplementary Figure 1 legend).

Fig 2. The quantitation of apoptosis requires normalisation to total cell (nuclear) number in the sections that are being compared. If total cell number was greater per section in the DM-NTD than DM or ND, then an increased apoptotic count would be expected, even if there was no difference in apoptosis incidence. The authors should either present data on average nuclear number per section for the three treatments (do they differ?), or actually calculate apoptotic indices (no. apoptotic nuclei/total no. nuclei). In addition, the graphs in (C) require a key to define the bar colors, or insert labels on the X-axis.

Response: We calculated apoptotic indices (no. apoptotic nuclei/total no. nuclei in the neuroepithelia) and new data was plotted in the graphs. The Y- and X-axis for the graphs in (now Fig. 1C) are now labeled.

Fig 3. The “green LC3-GFP puncta” which are taken as evidence of autophagic activity are not visible in these low magnification images of whole embryo sections. Please insert some specimen high magnification images to show the puncta clearly, or refer to Fig 5 where they are shown.

Response: In the figure legend (now Figure 2), we refer to Fig. 4 (Fig. 5 in original submission) for high magnification images in visualizing green LC3-GFP puncta.

Fig 4. Although the statistical comparison data for (C) are shown separately in Suppl Table 1, the statistically significant differences should also be indicated on the graphs, to make the result easier for the reader to appreciate. Also applies to Fig 9H.

Response: Statistically significant differences are indicated in Fig. 3C (originally Fig. 4C) and Fig. 8H (originally Fig. 9C).

Fig 5. It would aid the reader if the statistical differences were shown more clearly on graphs. Placing an asterisk over one bar does not indicate which other bar(s) it was compared with. Please insert tie-lines to link bars that were compared statistically, with overlying asterisks. Same comment applies to Figs 2C and Figs 6-10.

Response: In bar graphs with only one group that is significantly different than

other groups, one asterisk may be sufficient to indicate the difference. In cases that a bar graph has multiple groups with statistical significance, we inserted tie-lines to link bars that were compared statistically, with overlying asterisks.

Fig 6. Please refer to the images of defective and normal mitochondria in Suppl Fig 1, to illustrate how the distinction was made for quantitation in this figure. In Suppl Fig 1, please indicate the normal and defective mitochondria by labels or arrows.

Response: In Fig. 5 (originally Fig. 6), we refer to the images of defective and normal mitochondria in Suppl Fig 2 (Originally Suppl Fig. 1). Arrows are used in indicating defective mitochondria in the Supplementary Figure 2 (originally supplementary Fig. 1).

Fig 7. "The dense blue V shape areas are the neural tubes." This description is not accurate, as at E8.75 this is neural plate that has not yet closed to form the neural tube. Please re-phrase and also indicate which level of the body axis (hindbrain?) the images are showing. Also applies to Figs 9A and 10F.

Response: We have made corrections in now Fig. 6E, 8A and 9F.

Fig 8, D-F. Did the authors consider looking at cells exposed to caPKC α or the miR-129-2 mimic in combination with high glucose? Is there an additive effect? Please label the Y-axis in (A) to clearly indicate that this is relative mRNA level, as already done in (F) and (H).

Response: We do not expect any additive effect in combination of caPKC α and the miR-129-2 mimic because miR-129-2 is a downstream effector of PKC α activation (Fig. 7K, L, M). The Y-axis in Fig. 8A (now Fig. 7A) is now clearly labeled.

Fig 9. The lettering on the construct in (A) is too small to read. Please enlarge.

Response: We have enlarged the letters (now is Fig. 8A).

--

Reviewer #4 (Remarks to the Author):

Comments on Wang et al

The paper by Wang et al. investigates the mechanisms of diabetes induced neuronal tube defects. In particular, the authors show that diabetes activates PKC α , which in turn increases the expression of micro-RNA 129-2. This micro-

RNA subsequently represses the expression of PGC-1alpha. Since PGC-1alpha is an activator of autophagy, this pathway is shown here to inhibit autophagy. The authors further show that overexpression of PGC-1alpha in the neuroepithelium during development ameliorates the NTD defects caused by diabetes. The manuscript therefore reports a strong correlation between PKCalpha activation, PGC-1alpha expression and autophagy, although it is not clear to which extent all beneficiary effects of PGC-1alpha expression are due to autophagy induction. In summary, the authors report a detailed and mechanistic study that will be interested for scientists working on diabetes, NTDs and autophagy.

Response: We greatly appreciate the reviewer's positive comments.

There are a number of comments and questions the authors should address:

Major points:

1. From Fig. 5B and 9D it is unclear which structures were considered as puncta for the quantifications shown to the right. It seems that there is generally less GFP-LC3 in DM-WT cells. The blots shown in Fig. 5C and to some extent also Fig. 9C also suggest that there is less LC3 in DM-WT cells. While the LC3 in Fig. 5C and 9C are the endogenous proteins and could be transcriptionally regulated, I understand that GFP-LC3 is a transgene and should therefore not be subject to the same regulation. Do the authors have an explanation for this?

Response: For quantification of the GFP-LC3 puncta, we have described in the "Experimental Procedures" section. Briefly GFP-LC3 punctate with a diameter greater than or equal to 20 pixels in each section was calculated. Thus, the images captured the aggregated GFP-LC3 (GFP-LC3 puncta) fluorescent signal that is much stronger than that of individual GFP-LC3 (*Molecular biology of the cell* 15, 1101-1111, (2004)). In neuroepithelial cells of DM embryos, individual GFP-LC3 protein was diffused in cytoplasm, didn't form puncta, and had a much lower fluorescent signal that was not captured in the images. Total GFP-LC3 transgene product in Fig. 5B and Fig. 9D (now Fig. 4B, Fig. 8D) should be at the similar level among different groups.

We have included the above information into the corresponding part of the "Experimental Procedures" section.

2. Related to the point above, the authors quantify the total amount of LC3-II in Fig. 5C and it seems clear that there is less LC3-II in DM-WT cells compared to Prkca^{-/-} DM cells. However, the total amount of LC3 is also lower in DM-WT cells. Can the authors provide a value for the ration of LC3-II/LC3-I?

Response: We have provided the value for the ration of LC3-II/LC3-I in Fig.5C and Fig.9C (now Fig. 4C and Fig. 8C). The new data still support our hypothesis that maternal diabetes reduces the ration of LC3-II/LC3-I.

3. Can the authors explain how cyto-ID staining works (Fig. 5D-F)? This information is important to interpret the results.

Response: We have detailed the Cyto-ID autophagy staining protocol in the "Experimental Procedures" section.

4. In Fig. 5F the authors count over 500 puncta and by implication autophagosomes/cell. This is a very high number, in particular as the quantification of E8.75 neuroepithelial cells yields values that are about 50-100 times lower (Fig. 5B). Are authors sure that their quantification methods count individual autophagosomes?

Response: There're several reasons causing that the autophagosome counting is higher in the *in vitro* experiments.

a) The GFP-LC3 puncta counted in neuroepithelial cell of E8.75 embryos were conducted on sections. Embryonic neuroepithelial cells are relatively small, with a diameter about 20 μm according to our measurement, and have little cytoplasm. The sections are 5 μm thick, which means that we only counted about $\frac{1}{4}$ of the puncta in one neuroepithelial cell. In contrast, the puncta counted in cultured cells are based on a whole cell, and the diameters of our cultured cells are 40-50 μm , so the whole area is 4-6 times larger than that of an embryonic cell. If we assume that the density of autophagosome is at the same level in different type of cells, then the number of puncta per cell in cultured cells should be 16-25 times higher than embryonic cells.

b) The experiment methods are different. Embryonic samples were fixed with PFA, then cryo-sectioned and observed under confocal microscope. Some of the GFP fluorescent signals may be lost during this tissue preparation process. In contrast, the cultured cells were observed as live cells after cyto-ID staining, with no or very little signal lost.

c) The Cyto-ID staining detects pre-autophagosomes, autophagosomes, and autolysosomes, whereas the GFP-LC3 puncta method may just only detect matured autophagosomes depending on the defined size of the puncta.

We have clarified these points in the "Experimental Procedures" section.

5. The co-localization of GFP-LC3 with MitoID shown in Fig. S4B is not very

convincing as the signals are very diffuse. Either the authors show better pictures or they tone down their statements.

Response: We added high magnification inserts for visualizing the co-localization of GFP-LC3 puncta with MitoID in Fig. S5B (Originally Fig. S4B).

6. The data for Fig. 3B should be quantified

Response: We have quantified the data in Fig. 3B.

7. Fig. 4A, B should be quantified.

Response: We have quantified the data in Fig. 4A.

8. Details about the fluorescent microscopy setting are missing. Were all pictures in a given Figure taken with the same settings?

Response: We have provided the information of fluorescent microscopy setting in the "Experimental Procedures" section. All pictures in a given Figure taken with the same setting.

9. For Fig. 5A and 9B the authors should indicate how many autophagosomes were counted in total.

Response: We have provided the information of autophagosome quantification in figure legends: EM pictures were taken with an electron microscope (model: Joel JEM-1200EX) under 12K resolution. Each image covered $9.73 \mu\text{m}^2$ areas. The numbers of autophagosome on each image were counted and divided by cell nuclei in that image to get the value of autophagosome per cell. Five images of each embryo and three embryos from different mothers were quantified for each group. The total numbers of autophagosomes counted are eighty-six for Fig. 5A (now Fig. 4A) and eighty-one for Fig. 9B (now Fig. 8B).

10. The authors write that 83% of NTDs were comprised of exencephaly (Page 7). How do the authors get to this number?

Response: We have corrected the ratio which should be 74.3% (the number of embryos with exencephaly divided by the number of total observed NTD embryos). Among the NTD embryos, 74.3% of them are exencephalic and the other 25.7% are other type of NTDs.

Minor points

11. The x-axis in Fig. 2C needs labels

Response: We have provided labels in Fig. 2C.

12. The second sentence of the "In Brief" section is unclear.

Response: We have rewritten this sentence.

Reviewer #1 (Remarks to the Author)

I have carefully read the authors' responses and found that my questions and comments have been responded to in a satisfactory manner.

I have no more critique.

I wish the authors good luck with the remaining reviewing process, and look forward to read their contribution in another format.

Reviewer #2 (Remarks to the Author)

Reviewer #3 (Remarks to the Author)

The authors have performed additional analysis and rewritten the manuscript, and have in consequence answered all my questions and suggestions in a satisfactory way. Nevertheless, on reading the revised version, I have noted several further problems with the manuscript that need to be addressed before publication.

Please remove or rewrite the following sentences: Lines 56-57: "Folate cannot prevent diabetic embryopathy" and lines 84-85: "Surprisingly, multivitamins with folic acid, the only preventive measure to NTDs in the general population, cannot effectively prevent maternal diabetes-induced NTDs". References 5 and 6 are cited to support these statements. However, the Abstract to reference 5 (Correa et al) says: "The lack of periconceptional use of vitamins or supplements that contain folic acid may be associated with an excess risk for birth defects due to diabetes mellitus". Moreover, reference 6 (Oakley) says: "The data in the article by Correa et al suggest that folic acid fortification has removed the excess risk of spina bifida and anencephaly among the children of women with preexisting diabetes mellitus". In other words, taking multivitamins containing folic acid does reduce the risk of birth defects, particularly NTDs. The authors' statements are therefore incorrect, and these sentences need to be removed or modified to make clear the actual relationship between folate supplements and NTDs in diabetic pregnancies.

Abstract, line 42. Should read: "Deleting the Prkca gene, which encodes [or codes for] PKC α "

Lines 92-94 - Please combine the two following two sentences which are repetitive: "Maternal diabetes suppresses autophagy in the developing neuroepithelium. However, it is unclear how maternal diabetes represses autophagy during neurulation in the developing neuroepithelium."

Line 122. "Microcephaly" is not usually considered among the NTDs. Please make clear that microcephaly was observed, in addition to NTDs.

Lines 198-200. It should be made clear that this sentence refers to an experiment involving cultured cells. The sentence should state which cell line was used. A similar comment applies to other sentences, e.g. lines 212-218. The reader should be in no doubt about whether data are being presented from embryos or from cultured cells.

Line 21. "Luciferase" is misspelt.

Lines 256-259. In these two sentences, please make clear that the 2.2% NTDs in PPARGC1A+ embryos and the 22.7% in WT embryos, is under diabetic conditions.

Line 264. Should say "defective mitochondria".

Lines 299-300. "These evidence" should be corrected.

Reviewer #4 (Remarks to the Author)

In general, the authors have addressed all my comments. One issue that remains is that the data in Fig. S5B showing co-localization of GFP-LC3 puncta with MitoID is not convincing.

Responses to Reviewers' Concerns

REVIEWERS' COMMENTS:

Reviewer #1 (Remarks to the Author):

I have carefully read the authors' responses and found that my questions and comments have been responded to in a satisfactory manner.

I have no more critique.

I wish the authors good luck with the remaining reviewing process, and look forward to read their contribution in another format.

--

Reviewer #2

<Editor note: was satisfied and had no further comments for you>

--

Reviewer #3 (Remarks to the Author):

The authors have performed additional analysis and rewritten the manuscript, and have in consequence answered all my questions and suggestions in a satisfactory way. Nevertheless, on reading the revised version, I have noted several further problems with the manuscript that need to be addressed before publication.

Please remove or rewrite the following sentences: Lines 56-57: "Folate cannot prevent diabetic embryopathy" and lines 84-85: "Surprisingly, multivitamins with folic acid, the only preventive measure to NTDs in the general population, cannot effectively prevent maternal diabetes-induced NTDs". References 5 and 6 are cited to support these statements. However, the Abstract to reference 5 (Correa et al)

says: “The lack of periconceptional use of vitamins or supplements that contain folic acid may be associated with an excess risk for birth defects due to diabetes mellitus”. Moreover, reference 6 (Oakley) says: “The data in the article by Correa et al suggest that folic acid fortification has removed the excess risk of spina bifida and anencephaly among the children of women with preexisting diabetes mellitus”. In other words, taking multivitamins containing folic acid does reduce the risk of birth defects, particularly NTDs. The authors’ statements are therefore incorrect, and these sentences need to be removed or modified to make clear the actual relationship between folate supplements and NTDs in diabetic pregnancies. Response: We acknowledged that our statements are incorrect, and have removed the corresponding sentences.

Abstract, line 42. Should read: “Deleting the Prkca gene, which encodes [or codes for] PKC α ”

Response: A change has been made.

Lines 92-94 - Please combine the two following two sentences which are repetitive: “Maternal diabetes suppresses autophagy in the developing neuroepithelium.

Response: The two sentences are combined to one.

Line 122. “Microcephaly” is not usually considered among the NTDs. Please make clear that microcephaly was observed, in addition to NTDs.

Response: We have made it clear that microcephaly was observed, in addition to NTDs.

Lines 198-200. It should be made clear that this sentence refers to an experiment involving cultured cells. The sentence should state which cell line was used. A similar comment applies to other sentences, e.g. lines 212-218. The reader should be in no doubt about whether data are being presented from embryos or from cultured cells.

Response: We have made it clear that these experiments were done in cultured C17.2 cells.

Line 210. “luciferase” is misspelt.

Response: We have corrected the typo.

Lines 256-259. In these two sentences, please make clear that the 2.2% NTDs in PPARGC1A+ embryos and the 22.7% in WT embryos, is under diabetic conditions.

Response: We have made it clear that these NTD rates were from diabetic conditions.

Line 264. Should say “defective mitochondria”.

Response: We have corrected the typo.

Lines 299-300. “These evidence” should be corrected.

Response: We have changed this expression into “These findings”.

--

Reviewer #4 (Remarks to the Author):

In general, the authors have addressed all my comments. One issue that remains is that the data in Fig. S5B showing co-localization of GFP-LC3 puncta with MitolD is not convincing.

Response: We have removed Supplementary Fig. 5B and corresponding description in the manuscript.